# A lncRNA identifies *Irf8* enhancer element in negative feedback control of dendritic cell differentiation

**Huaming Xu[1,2], Zhijian Li[3†], Chao-Chung Kuo[3‡], Katrin Götz[1,2§], Thomas Look[1,2#], Marcelo AS de Toledo[1,2,4], Kristin Seré[1,2§], Ivan G Costa[3], Martin Zenke[1,2,4*]**

[1]Department of Cell Biology, Institute of Biomedical Engineering, RWTH Aachen University Medical School, Aachen, Germany; [2]Helmholtz Institute for Biomedical Engineering, RWTH Aachen University, Aachen, Germany; [3]Institute for Computational Genomics, RWTH Aachen University Medical School, Aachen, Germany; [4]Department of Hematology, Oncology, Hemostaseology, and Stem Cell Transplantation, Faculty of Medicine, RWTH Aachen University, Aachen, Germany

**\*For correspondence:**
martin.zenke@rwth-aachen.de

**Present address:** [†]Broad Institute of MIT and Harvard, Cambridge, Massachusetts, United States; [‡]Interdisciplinary Centre for Clinical Research (IZKF) Aachen, RWTH Aachen University Medical School, Aachen, Germany; [§]Department of Cell and Tumor Biology, RWTH Aachen University Medical School, Aachen, Germany; [#]Laboratory for Molecular Neuro-Oncology, Department of Neurology, University Hospital Zurich and University of Zurich, Zurich, Switzerland

**Competing interest:** The authors declare that no competing interests exist.

**Abstract** Transcription factors play a determining role in lineage commitment and cell differentiation. Interferon regulatory factor 8 (IRF8) is a lineage determining transcription factor in hematopoiesis and master regulator of dendritic cells (DC), an important immune cell for immunity and tolerance. IRF8 is prominently upregulated in DC development by autoactivation and controls both DC differentiation and function. However, it is unclear how *Irf8* autoactivation is controlled and eventually limited. Here, we identified a novel long non-coding RNA transcribed from the +32 kb enhancer downstream of *Irf8* transcription start site and expressed specifically in mouse plasmacytoid DC (pDC), referred to as *lncIrf8*. The *lncIrf8* locus interacts with the *Irf8* promoter and shows differential epigenetic signatures in pDC versus classical DC type 1 (cDC1). Interestingly, a sequence element of the *lncIrf8* promoter, but not *lncIrf8* itself, is crucial for mouse pDC and cDC1 differentiation, and this sequence element confers feedback inhibition of *Irf8* expression. Taken together, in DC development *Irf8* autoactivation is first initiated by flanking enhancers and then second controlled by feedback inhibition through the *lncIrf8* promoter element in the +32 kb enhancer. Our work reveals a previously unrecognized negative feedback loop of *Irf8* that orchestrates its own expression and thereby controls DC differentiation.

## Editor's evaluation

Authors provide valuable evidence identifying a lncRNA transcribed specifically in the pDC subtype from the +32Kb promoter region which is also the region for the enhancer for Irf8 specifically in the cDC1 subtype. With convincing methodology, they provide in-depth analysis about the possible role of lncIrf8, and its promoter region and cross-talk with Irf8 promoter to identify that it is not the lncIRF8 itself but its promoter region that is crucial for pDC and cDC1 differentiation conferring feedback inhibition of Irf8 transcription. The work will be of interest to immunologists working on immune cell development.

## Introduction

Lineage-determining transcription factors (TF) are master regulators of gene programs that frequently initiate self-reinforcing loops by autoactivation. TF autoactivation is important for cells to pass restriction points during development (referred to as points of no return) and to enforce cellular identity.

Molecular circuitries of autoactivation have been studied for several TF, such as GATA-binding factor 1 (GATA1), PU.1 (encoded by Spi1), CCAAT enhancer-binding protein α and ε (C/EBPα and ε; *Graf and Enver, 2009*; *Loughran et al., 2020*; *Nishimura et al., 2000*; *Okuno et al., 2005*; *Theilgaard-Mönch et al., 2022*). A further example is interferon regulatory factor 8 (IRF8), which shows autoactivation in cooperation with basic leucine zipper ATF-like transcription factor 3 (BATF3; *Anderson et al., 2021*; *Grajales-Reyes et al., 2015*). An important principle in nature is negative feedback control to avoid signal overshooting and toxicity. Negative feedback control applies also to lineage-determining TF; however, there is a paucity on our knowledge of the molecular mechanisms involved.

IRF8 is a hematopoietic TF positioned at the center of the regulatory gene network for dendritic cell (DC) development (*Anderson et al., 2021*; *Belz and Nutt, 2012*; *Chauvistré and Seré, 2020*; *Kim et al., 2020*; *Lin et al., 2015*; *Nutt and Chopin, 2020*; *Tamura et al., 2015*; *Verlander et al., 2022*). IRF8 is a member of the interferon regulatory factor (IRF) family of TF. Initially members of this TF family were found to mediate the induction of interferon induced genes, but are now known to serve diverse functions in regulating the immune system (*Honda and Taniguchi, 2006*; *Tamura et al., 2008*). *Irf8* knockout mice show abnormal development of classical DC type 1 (cDC1) and plasmacytoid DC (pDC) (*Durai et al., 2019*; *Schiavoni et al., 2002*; *Sichien et al., 2016*; *Tsujimura et al., 2003*). *Irf8* is prominently upregulated during DC development by autoactivation (*Grajales-Reyes et al., 2015*; *Lin et al., 2015*), yet how *Irf8* autoactivation is controlled and eventually limited, and the epigenetic mechanisms involved is largely unknown.

*Irf8* expression in hematopoietic cells is induced and maintained by enhancers located at –50 kb,+32 kb,+41 kb and +56 kb relative to *Irf8* transcription start site (TSS) (*Anderson et al., 2021*; *Bagadia et al., 2019*; *Durai et al., 2019*; *Grajales-Reyes et al., 2015*; *Murakami et al., 2021*; *Schönheit et al., 2013*). Enhancers are *cis*-regulatory sequences with multiple TF binding sites that cooperatively bind TF and thereby activate transcription, as demonstrated by many studies including our work (*Davidson et al., 1986*; *Long et al., 2016*; *Wildeman et al., 1986*; *Zenke et al., 1986*). Enhancers regulate complex gene networks and can also produce non-coding RNA, referred to as enhancer RNA (eRNA). eRNA serve as an indicator for enhancer activity and some eRNA have an activity on their own and act in cis or trans to regulate cell fate decisions (*Sartorelli and Lauberth, 2020*; *Statello et al., 2021*). Enhancer-associated long non-coding RNA (lncRNA) represent a class of lncRNA transcribed from active enhancers. Thus, eRNA and enhancer-associated lncRNA provide opportunities to detect enhancer activity and to investigate enhancer function.

DC are highly specialized immune cells that play a critical role in regulating innate and adaptive immune responses (*Cabeza-Cabrerizo et al., 2021*). DC develop from hematopoietic stem cells (HSC) via successive steps of lineage commitment and differentiation. More specifically, HSC develop into multipotent progenitors (MPP) that are committed to DC restricted common DC progenitors (CDP) and differentiate into classic DC (cDC) type 1 and type 2 (cDC1 and cDC2, respectively) and pDC (*Anderson et al., 2021*; *Cabeza-Cabrerizo et al., 2021*; *Ginhoux et al., 2022*; *Nutt and Chopin, 2020*; *Rodrigues and Tussiwand, 2020*). pDC were recently also shown to develop from lymphoid progenitors (*Dress et al., 2019*; *Rodrigues et al., 2018*; *Rodrigues and Tussiwand, 2020*). Differential expression of *Irf8* regulates DC and monocyte specification in a dose-dependent manner (*Cytlak et al., 2020*; *Murakami et al., 2021*). *Irf8* expression starts at the CDP stage, and is high in pDC and cDC1, which is attributed to the autoactivation of *Irf8* during DC subsets specification (*Grajales-Reyes et al., 2015*; *Lin et al., 2015*). Interestingly, IRF8 can act as a transcriptional activator or repressor in hematopoiesis by interacting with different partner TF and binding to specific DNA sequences (*Tamura et al., 2015*).

As an activator, IRF8 binds to its own promoter in DC differentiation, which is considered as the autoactivation capacity of *Irf8* (*Grajales-Reyes et al., 2015*; *Lin et al., 2015*). For instance, IRF8 interacts with partner TF, such as PU.1, to initiate *Irf8* autoactivation at the CDP stage (*Grajales-Reyes et al., 2015*). Inversely, IRF8 inhibits C/EBPα activity in neutrophil differentiation (*Kurotaki et al., 2014*). IRF8 also represses C/EBPβ to generate and maintain DC lineage-specific enhancer landscapes (*Bornstein et al., 2014*). In addition, IRF8 is important for the *Myc-Mycl* transition in DC differentiation (*Anderson III et al., 2021*). IRF8 represses *Myc* expression in progenitors, while IRF8 at high levels interacts with PU.1 and drives *Mycl* expression (*Anderson III et al., 2021*). All this emphasizes the central position of IRF8 in coordinating the gene network that regulates DC differentiation and function.

During DC differentiation, the *Irf8* gene locus shows high epigenetic dynamics, including histone modifications and TF binding identified by ChIP-seq (*Chauvistré and Seré, 2020*; *Durai et al., 2019*; *Grajales-Reyes et al., 2015*; *Lin et al., 2015*), chromatin accessibility measured by ATAC-seq (*Kurotaki et al., 2019*; *Li et al., 2019*), and three-dimensional chromatin structure remodeling determined by chromosome conformation capture (3 C) (*Kurotaki et al., 2022*; *Schönheit et al., 2013*). All this emphasizes the impact of epigenetic regulators on *Irf8* gene activity in DC differentiation. Notably, *Irf8* is flanked by multiple enhancers at –50 kb,+32 kb,+41 kb, and +56 kb that regulate *Irf8* expression in hematopoietic cells (*Anderson et al., 2021*; *Murakami et al., 2021*). These four enhancers were found to be driven by PU.1, BATF3, E proteins and Runt-related transcription factor (RUNX)-core binding factor beta (CBFβ) (RUNX-CBFβ), respectively (*Bagadia et al., 2019*; *Durai et al., 2019*; *Grajales-Reyes et al., 2015*; *Murakami et al., 2021*; *Schönheit et al., 2013*).

Chromatin conformation, particularly enhancer promoter interactions, provides a platform for TF-driven gene regulation and serves as a driving force for cell-fate determinations (*Misteli and Finn, 2021*; *Oudelaar and Higgs, 2021*; *Stadhouders et al., 2019*). Schönheit et al. demonstrated *Irf8* promoter interactions with its upstream enhancers by quantitative 3 C (*Schönheit et al., 2013*). In this study PU.1 was found to regulate chromatin remodeling between the –50 kb enhancer and the *Irf8* promoter in myeloid differentiation. In a recent study *Kurotaki et al., 2022* determined the higher-order chromatin structure in DC progenitors, cDC1 and cDC2 on a genome-wide scale by Hi-C. In this study, reorganization of chromatin conformation at DC-specific gene loci was observed during cDC differentiation, and IRF8 was found to promote chromatin activation in DC progenitors leading to cDC lineage-specific gene expression. However, high resolution maps of the physical chromatin interactions of the *Irf8* promoter with upstream and downstream enhancers in the full complement of DC subsets, including pDC, are required for understanding *Irf8* regulation during DC differentiation.

Frequently, chromatin data, including ATAC-seq and/or ChIP-seq data, are used to identify regulatory elements of gene transcription. Here we embarked on a different approach and searched for lncRNA, which by themselves might have regulatory functions or are indicative of enhancer activity. We identified a novel lncRNA transcribed from the *Irf8* +32 kb enhancer, which is specifically expressed in pDC, referred to as *lncIrf8*. We found that the *lncIrf8* promoter element but not *lncIrf8* itself impacts pDC and cDC1 development. Thus, *lncIrf8* acts as an indicator for the *Irf8* +32 kb enhancer activity. Importantly, our study revealed a previously unrecognized negative feedback loop of *Irf8* in DC differentiation. *Irf8* first activates its expression by autoactivation via the +32 kb enhancer and second limits its own expression through the *lncIrf8* promoter element in the +32 kb enhancer.

## Results

### *lncIrf8* marks a pDC-specific *Irf8* enhancer element

*Irf8* expression in DC development is subject to complex epigenetic regulation. Here, we used an integrated approach with RNA-seq, ATAC-seq, ChIP-seq and Capture-C to track the dynamics of gene expression, histone modification and chromatin conformation in the sequel MPP, CDP, pDC, cDC1, and cDC2 (*Figure 1*, *Figure 1—figure supplements 1 and 2*).

We performed de novo transcript assembly of the RNA-seq data and detected two previously unknown transcripts without coding potential downstream of *Irf8*: a pDC specific lncRNA (*Tcons_00190250*) in the following referred to as *lncIrf8* and a cDC1 specific lncRNA (*Tcons_00190258*; *Figure 1A* and *Figure 1—figure supplement 1*). *lncIrf8* and *Tcons_00190258* show the same expression pattern in pDC and cDC1, respectively, in BM and spleen (*Figure 1—figure supplement 3*), as revealed by reanalyzing scRNA-seq and bulk RNA-seq data (*Pang et al., 2022*; *Rodrigues et al., 2018*). *lncIrf8* is transcribed within an enhancer region located 32 kb downstream of the *Irf8* TSS labeled by H3K27ac and H3K4me1 and occupied by DC differentiation-associated TF, such as IRF8 and PU.1 (*Figure 1A* and *Figure 1—figure supplement 1*). This region is largely devoid of H3K9me3, a chromatin modification frequently associated with heterochromatin, indicating an open chromatin configureuration in DC (*Figure 1—figure supplement 1*). In addition, sequences of this region have been implicated in DC development and referred to as +32 kb enhancer (*Durai et al., 2019*). Thus, we proceeded to study *lncIrf8* in detail.

ATAC-seq analysis revealed further details of the *lncIrf8* region in CDP, pDC, cDC1 and cDC2 (*Figure 1A*, *Figure 2A* and *Figure 1—figure supplement 1*). In cDC1 the prominent ATAC-seq and

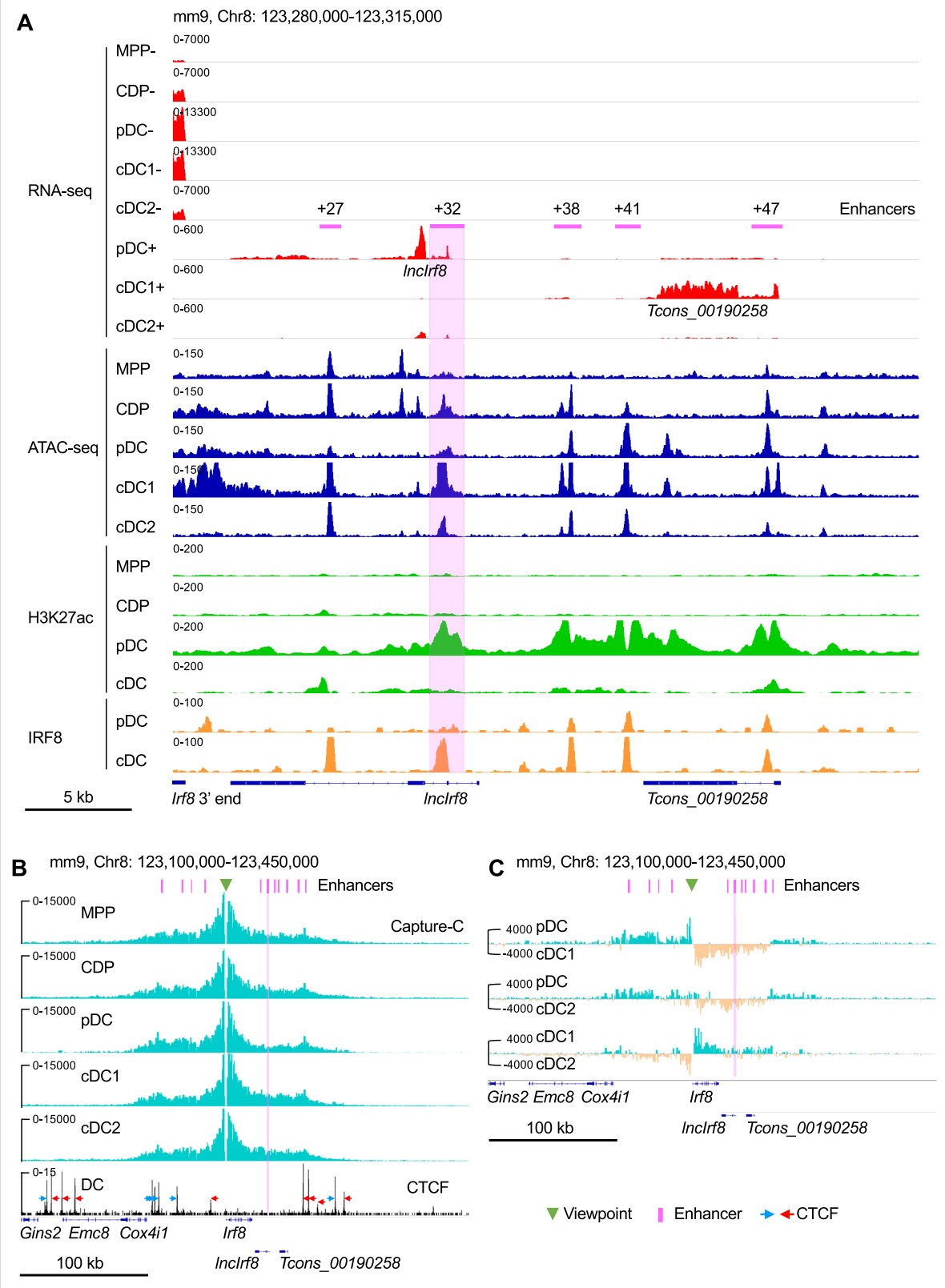

**Figure 1.** *Irf8* epigenetic signatures and promoter-enhancer interaction maps during DC differentiation. (**A**) Gene expression and epigenetic signatures of *Irf8* downstream region in MPP, CDP, pDC, all cDC, cDC1, and cDC2 are visualized by IGV browser. Gene expression was measured by RNA-seq, chromatin accessibility by ATAC-seq, H3K27ac and IRF8 binding by ChIP-seq. Positions of *Irf8* 3' end, *Irf8* enhancers, pDC specific *lncIrf8* and cDC1 specific *Tcons_00190258* lncRNA are indicated. For RNA-seq - and +strands are shown. Scale bar: 5 kb. (**B**) Physical interactions of *Irf8* promoter with

*Figure 1 continued on next page*

*Figure 1 continued*

flanking sequences in MPP, CDP, pDC, cDC1, and cDC2 by nuclear-titrated (NuTi) Capture-C (turquoise), and CTCF binding by ChIP-seq in DC (*Garber et al., 2012*). Mean numbers of unique interactions normalized to a 300 kb region around the *Irf8* promoter viewpoint (green triangle) and scaled by a factor of 1,000,000 are shown (n=2). The orientations of CTCF binding are indicated with blue and red arrows. *Tcons_00190258* refers to the cDC1-specific lncRNA shown in (**A**). Scale bar: 100 kb. (**C**) Comparations of the chromatin interactions with *Irf8* promoter in pDC, cDC1 and cDC2. Differential tracks were created by subtraction of the mean normalized tracks of (**B**). Pairwise comparisons are shown and color coded. Turquoise and orange tracks represent specific interactions with the *Irf8* promoter in the indicated cell types. Scale bar: 100 kb. Purple bars and lines indicate the position of flanking enhancers relative to *Irf8* TSS. The purple bars from left to right represent –50 kb, –34 kb, –26 kb, –16 kb,+27 kb,+32 kb,+38 kb,+41 kb,+47 kb,+56 kb and +62 kb enhancer, respectively (panels B and C). *Irf8* +32 kb enhancer is highlighted by purple box.

The online version of this article includes the following figure supplement(s) for figure 1:

**Figure supplement 1.** Epigenetic signatures of *Irf8* locus during DC differentiation.

**Figure supplement 2.** In vitro DC differentiation of HoxB8 MPP and nuclear-titrated (NuTi) Capture-C.

**Figure supplement 3.** pDC specific *lncIrf8* and cDC1 specific *Tcons_00190258* lncRNA expression in ex-vivo DC.

IRF8 peaks mark the cDC1 specific +32 kb enhancer (*Durai et al., 2019*). In pDC the ATAC-seq peak is smaller and shifted further towards downstream but aligns well with the valley in the prominent H3K27ac peak. This ATAC-seq peak marks the *lncIrf8* promoter and aligns with p300 (*Durai et al., 2019*) and H3K4me3 (*Figure 2A* and *Figure 1—figure supplement 1*). All this indicates that this chromatin region is open and transcriptionally active in pDC, enabling *lncIrf8* transcription.

Next, we determined the chromatin conformation of the *Irf8* locus and the *lncIrf8* region. We generated interaction profiles by nuclear-titrated (NuTi) Capture-C in MPP, CDP, pDC, cDC1, and cDC2 (*Figure 1—figure supplement 2A and B*) using *Irf8* promoter as viewpoint. The *Irf8* promoter shows multiple interactions with regions spanning up to ~100 kb upstream and downstream of *Irf8* (*Figure 1B* and *Figure 1—figure supplement 1*). In pDC, the *Irf8* promoter interactions are stronger with the upstream sequences than with downstream sequences (*Figure 1C* and *Figure 1—figure supplement 2C*). In cDC1 *Irf8* promoter interactions are more confined to the regions downstream of *Irf8* compared to MPP, CDP and pDC (*Figure 1C* and *Figure 1—figure supplement 2C*). This suggests that upstream and downstream sequences of *Irf8* gene are involved in differentially regulating *Irf8* expression and controlling the development of pDC and cDC1, respectively.

The CCCTC-binding factor (CTCF) is important for regulation of chromatin conformation through loop extrusion (*Sanborn et al., 2015*) and we therefore visualized CTCF binding sites in the *Irf8* locus in DC (*Garber et al., 2012*). Interestingly, most of the *Irf8* flanking enhancers (*Durai et al., 2019*; *Grajales-Reyes et al., 2015*; *Murakami et al., 2021*; *Schönheit et al., 2013*) are located within convergent CTCF binding sites upstream and downstream of the *Irf8* gene (*Figure 1B* and *Figure 1—figure supplement 1*). There are also multiple interactions within this region without convergent CTCF binding sites, suggesting interactions with *Irf8* promoter in a CTCF independent manner, such as by TF binding, active histone modifications and gene transcription (*Figure 1B* and *Figure 1—figure supplement 1*; *Owens et al., 2022*).

Surprisingly, in pDC H3K27ac at the *lncIrf8* promoter is high, but this locus shows less interactions with the *Irf8* promoter in pDC compared to CDP, cDC1 and cDC2 (*Figure 1C* and *Figure 1—figure supplement 2C*). In addition, in pDC the IRF8 protein occupancy at the *lncIrf8* promoter is low and much higher in cDC (*Figure 1A* and *Figure 1—figure supplement 1*; *Durai et al., 2019*; *Grajales-Reyes et al., 2015*).

These observations warrant further studies and we thus proceeded to investigate the *lncIrf8* locus in detail.

## *lncIrf8* promoter KO compromises pDC and cDC1 development

First, we annotated *lncIrf8*. Our de-novo transcript assembly of RNA-seq data revealed different isoforms of *lncIrf8*, with the most prominent isoform comprising exon 2 and 3 (*Figure 1A*, *Figure 2A* and *Figure 1—figure supplement 1*). Additionally, 3' end and 5' end RACE PCR confirmed the anatomy of this *lncIrf8* isoform: two exons, one intron, and a polyA tail (*Figure 2A*). As expected *lncIrf8* is not conserved across species (data not shown), which is in line with the general characteristics of lncRNA. Then second, we deleted 300 bp in the *lncIrf8* promoter by CRISPR/Cas9 editing in conditionally immortalized HoxB8 MPP (*Figure 2A* and *Figure 2—figure supplement 1A–D*). The *lncIrf8* promoter is located in the *Irf8* +32 kb enhancer region and is in close proximity to the cDC1

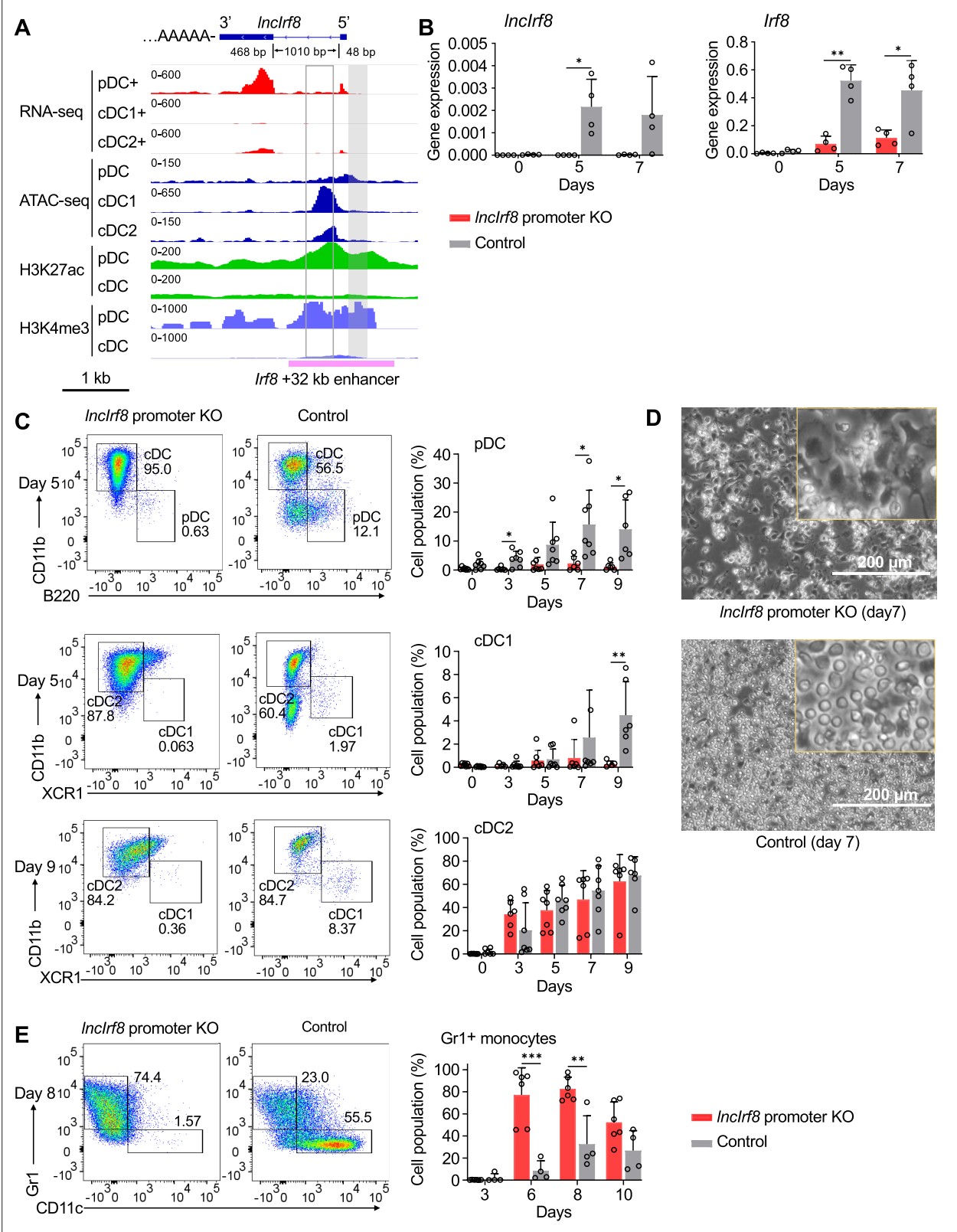

**Figure 2.** *lncIrf8* promoter KO compromises pDC and cDC1 development in vitro. (**A**) Genomic anatomy of *lncIrf8* locus determined by 3' and 5' RACE PCR. Blue box, exon 2 and 3 (48 bp and 468 bp, respectively). The 1010 bp intron and polyA tail are shown. Data of RNA-seq, ATAC-seq, ChIP-seq of H3K27ac (enhancer mark) and H3K4me3 (active promoter mark, near TSS) are visualized by IGV browser for the indicated cell populations (pDC, all cDC, cDC1 and cDC2). Grey box, *lncIrf8* promoter KO region; open box, cDC1 specific +32 kb enhancer by **Durai et al., 2019**. *Irf8* +32 kb enhancer based

*Figure 2 continued on next page*

*Figure 2 continued*

on the H3K27ac enhancer mark is indicated with a purple line. Scale bar: 1 kb. (**B**) Gene expression of *lncIrf8* and *Irf8* in *lncIrf8* promoter KO and control at day 0, 5, and 7 of Flt3L directed DC differentiation. Gene expression was determined by RT-qPCR and normalized to *GAPDH*. n=4. (**C**) Representative flow cytometry analysis of Flt3L directed DC differentiation of *lncIrf8* promoter KO HoxB8 MPP and control (*Lutz et al., 2022*; *Xu et al., 2022*). pDC, all cDC, cDC1, and cDC2 were gated as in *Figure 2—figure supplement 1E* and are shown. Bar diagrams depict quantification of pDC, cDC1 and cDC2 normalized to living single cells on DC differentiation day 0, 3, 5, 7, and 9. n=6–7. (**D**) Representative phase-contrast microscopy images of *lncIrf8* promoter KO HoxB8 MPP and control on day 7 of Flt3L directed DC differentiation. Scale bar: 200 µm. (**E**) Representative flow cytometry analysis of spontaneous DC differentiation of *lncIrf8* promoter KO HoxB8 MPP and control with growth factors but without E2 (*Lutz et al., 2022*; *Xu et al., 2022*) at day 8. Gr1$^+$ monocytes and CD11c$^+$ DC are shown. Quantification of Gr1$^+$ monocytes of living single cells on day 3, 6, 8, and 10 of spontaneous DC differentiation. n=6, *lncIrf8* promoter KO; n=4, control. Empty gRNA vector or non-targeting gRNA vector HoxB8 MPP were used as controls. Data represent mean ± SD of at least three independent experiments with different HoxB8 MPP clones of *lncIrf8* promoter KO and control without deletion. *p<0.05, **p<0.01, ***p<0.001, multiple t-tests. Data that have no difference (p>0.05) are not labeled.

The online version of this article includes the following source data and figure supplement(s) for figure 2:

**Figure supplement 1.** Generating *lncIrf8* promoter KO HoxB8 MPP and its impact on DC differentiation.

**Figure supplement 1—source data 1.** Full raw unedited gel of genotyping of *lncIrf8* promoter KO HoxB8 MPP clones in *Figure 2—figure supplement 1C*.

**Figure supplement 1—source data 2.** Labeled uncropped gel of genotyping of *lncIrf8* promoter KO HoxB8 MPP clones in *Figure 2—figure supplement 1C*.

**Figure supplement 2.** Information of *lncIrf8* promoter KO.

**Figure supplement 3.** Impact of *lncIrf8* promoter KO on spontaneous DC differentiation.

**Figure supplement 4.** cDC1 specific +32 kb enhancer KO compromises cDC1 development in vitro.

**Figure supplement 4—source data 1.** Full raw unedited gel of genotyping of cDC1 specific +32 kb enhancer KO clones in *Figure 2—figure supplement 4B*.

**Figure supplement 4—source data 2.** Labeled uncropped gel of genotyping of cDC1 specific +32 kb enhancer KO clones in *Figure 2—figure supplement 4B*.

specific +32 kb enhancer (*Durai et al., 2019*; *Figure 2A* and *Figure 2—figure supplement 2*). The 300 bp deletion comprises the H3K4me3 promoter mark and is confined to open chromatin identified by ATAC-seq and positioned in the valley of the H3K27ac mark (*Figure 2A*). Additionally, it contains binding sites for IRF8, PU.1, and BATF3 TF, which are important for DC development (*Figure 2A* and *Figure 2—figure supplement 2B*).

Generation of a precise deletion requires clonal cell populations, which is hardly achieved in somatic cells due to their limited lifespan. Therefore, we developed a Mx-Cas9-GFP system of conditionally immortalized HoxB8 MPP, which upon differentiation faithfully recapitulate DC development (*Figure 2—figure supplement 1A, B*; *Xu et al., 2022*). HoxB8 MPP were obtained from bone marrow of Mx-Cas9-GFP mice by infection with the estrogen (E2) inducible HoxB8-ER. These HoxB9 MPP exhibited an extended lifespan and robust clonogenic potential and differentiated into all DC subsets in vitro and in vivo (*Xu et al., 2022*). Infection of gRNA targeting *lncIrf8* promoter in Mx-Cas9-GFP HoxB8 MPP and induction of Cas9 by interferon generated single-cell *lncIrf8* promoter KO clones. Five out of 71 single-cell colonies with homozygous deletions were further studied and subjected to DC differentiation (*Figure 2C–E*, *Figure 2—figure supplement 1C–G* and *Figure 2—figure supplement 3*).

*lncIrf8* promoter KO abolished *lncIrf8* expression during DC differentiation compared to control without deletion (*Figure 2B*). Surprisingly, *Irf8* expression was also severely compromised, which points to a cross-talk of the *lncIrf8* promoter element with the *Irf8* promoter. To determine whether *lncIrf8* promoter KO also impacts DC subsets, CD11c$^+$ DC, pDC and cDC subsets cDC1 and cDC2 were analyzed (*Figure 2C* and *Figure 2—figure supplement 1E–G*). Frequencies of pDC and cDC1 were severely reduced, while cDC2 were unaffected (*Figure 2C*). Accordingly, *lncIrf8* promoter KO cultures contained mainly cDC2 and some undifferentiated cells and were more homogenous compared to control without deletion, which contain multiple DC subsets (*Figure 2C and D*, *Figure 2—figure supplement 1F, G*).

*lncIrf8* promoter KO affected also the differentiation propensity of progenitors upon E2 withdrawal from MPP/CDP culture (*Figure 1—figure supplement 2A* and *Figure 2—figure supplement 3*). *lncIrf8* promoter KO showed a marked increase in strongly adhesive cells compared to control

(*Figure 2—figure supplement 3B*). *lncIrf8* promoter KO cultures had higher frequencies of Gr1 [+] monocytes (*Figure 2E*, *Figure 2—figure supplement 3I, J*) and lower frequencies of all DC subsets CD11c[+] DC, pDC, cDC1, and cDC2 (*Figure 2—figure supplement 3C–G*) compared to control without deletion.

The *lncIrf8* promoter element is in close proximity to the cDC1 specific +32 kb enhancer previously described in mice by *Durai et al., 2019* and thus we generated the cDC1 specific +32 kb enhancer KO following the same procedure as for the *lncIrf8* promoter KO. Five out of 165 single-cell clones with homozygous deletions of cDC1 +32 kb enhancer were subjected to DC differentiation and further analyzed (*Figure 2—figure supplement 4*). Similar to *lncIrf8* promoter KO, cDC1 specific +32 kb enhancer KO abolished *lncIrf8* expression and also decreased *Irf8* expression during DC differentiation (*Figure 2—figure supplement 4D*). The cDC1 +32 kb enhancer KO also compromised the frequency of cDC1 in Flt3L directed DC differentiation, while cDC2 were unaffected (*Figure 2—figure supplement 4E*). Frequencies of pDC were also compromised at day 7; however, this was not statistically significant. These observations are in line with previous studies in mice that cDC1 +32 kb enhancer KO compromised cDC1 differentiation and left pDC and cDC2 largely unaffected (*Durai et al., 2019*; *Murakami et al., 2021*). In addition, cDC1 +32 kb enhancer KO cultures had higher frequencies of Gr1 [+] monocytes upon spontaneous DC differentiation by withdrawal of E2 (*Figure 2—figure supplement 4F*) and thus showed a similar phenotype as the *lncIrf8* promoter KO upon spontaneous DC differentiation (*Figure 2E*).

Given the novel phenotype of the *lncIrf8* promoter KO, we proceeded to investigate the impact of the *lncIrf8* promoter KO on DC differentiation in vivo in mice. We transplanted CD45.2 *lncIrf8* promoter KO and CD45.2 control HoxB8 MPP into irradiated CD45.1 recipient mice (*Figure 3—figure supplement 1A*). DC in bone marrow and spleen were analyzed by flow cytometry on day 7 and 14 after cell transplantation (*Figure 3*, *Figure 3—figure supplement 1A, B*). In bone marrow, *lncIrf8* promoter KO cells mostly differentiated into Gr1 [+] monocytes, and lower frequencies of all DC subsets were observed on day 7 for *lncIrf8* promoter KO cells compared to control (*Figure 3A–F*). In spleen, frequencies of cell populations from *lncIrf8* promoter KO and control were similar to bone marrow, including lower frequencies of all DC subsets for *lncIrf8* promoter KO (*Figure 3G–L*). CD45.2 donor HoxB8 cells were largely lost at day 14 after cell transplantation (*Figure 3B–F* and *Figure 3H–L*).

Thus, *lncIrf8* promoter KO compromised pDC and cDC1 development both in vitro and in vivo.

## *lncIrf8* acts as an indicator of *Irf8* +32 kb enhancer activity in pDC

Knockout of *lncIrf8* promoter and thus abolishment of *lncIrf8* expression severely diminished pDC and cDC1 development in vitro and in vivo. The *lncIrf8* promoter is located within *Irf8* +32 kb enhancer (*Durai et al., 2019*) and thus it was important to determine whether *lncIrf8* itself plays a role in regulating pDC and cDC1 development. To address this question, we (i) overexpressed *lncIrf8* in wild-type MPP and (ii) re-expressed *lncIrf8* in *lncIrf8* promoter KO MPP and monitored its impact on DC development (*Figure 4* and *Figure 4—figure supplement 1*).

*lncIrf8* cDNA was cloned into a polyA lentivirus vector. An 'AATAAA' stop signal (*Alvarez-Dominguez et al., 2015*) was inserted at the 3' end of *lncIrf8* to avoid longer transcripts than *lncIrf8* (p*lncIrf8*-pA, *Figure 4B*). The respective pGFP-pA vector was used as control. *lncIrf8* overexpressing single-cell clones were generated by limiting dilution (*Figure 4—figure supplement 1A*), expanded and subjected to DC differentiation. As expected *lncIrf8* expression was markedly increased in p*lncIrf8*-pA infected cells compared to control, while there were no significant differences in *Irf8* expression between the two groups during DC differentiation (*Figure 4C*). Further, there were no differences in the frequencies of pDC, cDC1, and cDC2 between p*lncIrf8*-pA infected cells and controls (*Figure 4D and E*, *Figure 4—figure supplement 1B–D*), indicating that *lncIrf8* overexpression has no effect on *Irf8* expression and DC differentiation.

To further extend this observation we performed a *lncIrf8* rescue in *lncIrf8* promoter KO MPP. *lncIrf8* was re-expressed in *lncIrf8* promoter KO MPP by lentiviral vector and cells were subjected to DC differentiation (*Figure 4—figure supplement 1E*). *lncIrf8* RNA was effectively expressed and cells differentiated in response to Flt3L (*Figure 4—figure supplement 1F*). Yet frequencies of pDC and cDC1 were very low to absent and not rescued by *lncIrf8* expression (*Figure 4—figure supplement 1G, H*).

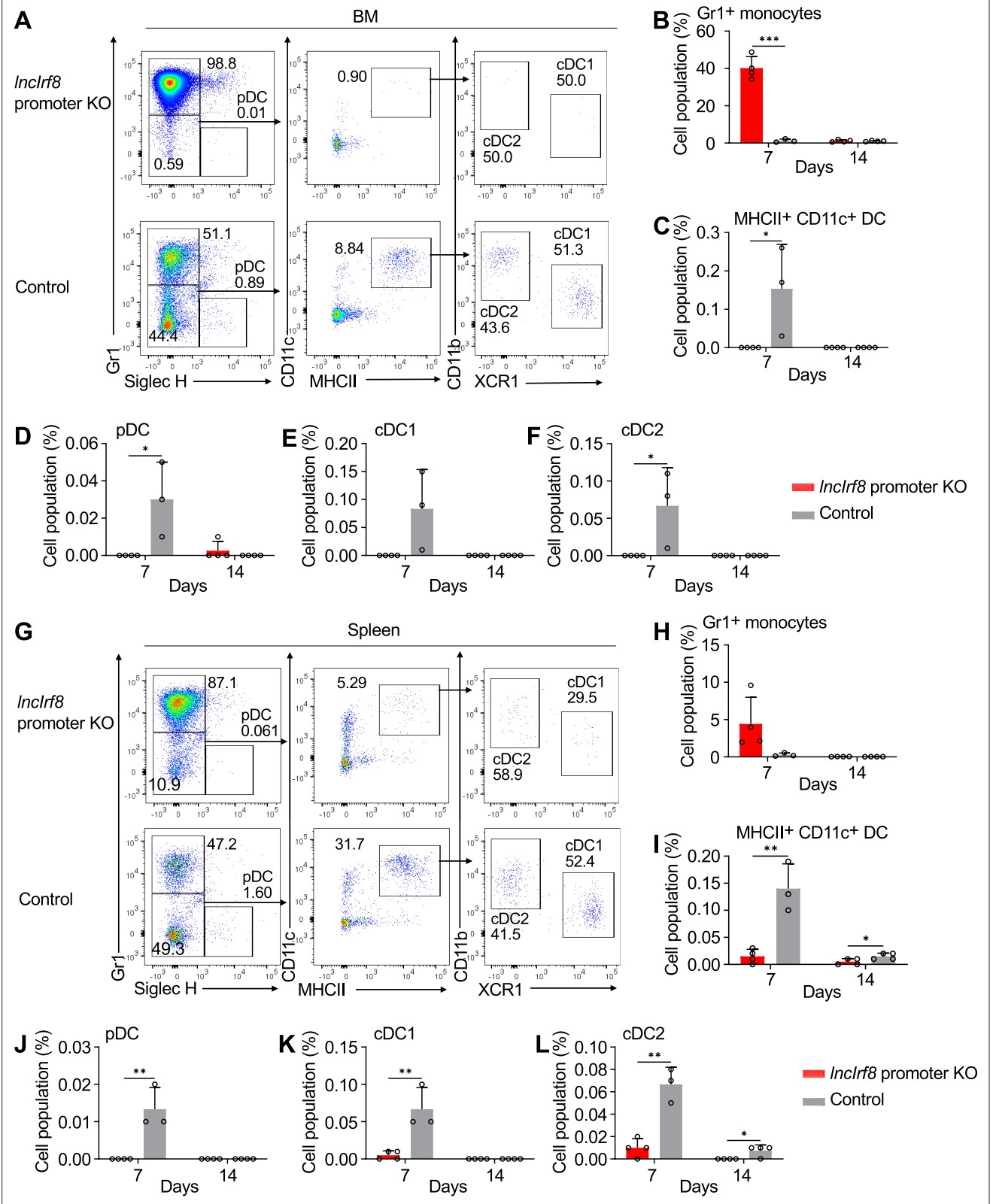

**Figure 3.** *lncIrf8* promoter KO comprises pDC and cDC1 development in vivo upon cell transplantation. (**A**) Representative flow cytometry analysis of CD45.2 *lncIrf8* promoter KO and control HoxB8 MPP in BM at day 7 after cell transplantation (for details see *Figure 3—figure supplement 1A, B*). Donor cell populations were gated from 7-AAD⁻ CD45.2⁺ Lin⁻ cells and Gr1 ⁺ monocytes, pDC, cDC1 and cDC2 are shown. (**B–F**) Quantification of Gr1 ⁺ monocytes, MHCII ⁺ CD11c⁺ DC, pDC, cDC1, and cDC2 of living single cells in BM on day 7 and 14 after cell transplantation (n=3–4). (**G**) Representative

*Figure 3 continued on next page*

*Figure 3 continued*

flow cytometry analysis of *lncIrf8* promoter KO and control HoxB8 MPP in spleen at day 7 after cell transplantation. Gating was as in panel (**A**). (**H–L**) Quantification of Gr1 $^+$ monocytes, MHCII $^+$ CD11c $^+$ DC, pDC, cDC1 and cDC2 on day 7 and 14 after cell transplantation (n=3–4). Data represent mean ± SD from 3 to 4 mice. *p<0.05, **p<0.01, ***p<0.001, multiple t-tests. Data that have no difference (p>0.05) are not labeled.

The online version of this article includes the following figure supplement(s) for figure 3:

**Figure supplement 1.** Cell transplantation of HoxB8 MPP.

In a nutshell, *lncIrf8* overexpression and rescue had no effects on pDC and cDC1 development. This strongly suggests that *lncIrf8* has no activity on its own in DC differentiation but rather acts as an indicator for the activation state of sequences within the *Irf8* +32 kb enhancer. In addition, the *lncIrf8* promoter comprises a sequence element with impact on pDC and cDC1 development.

## Activation of *lncIrf8* promoter promotes cDC1 development

Next, we proceeded to study the impact of the sequence element within *lncIrf8* promoter on *Irf8* expression and DC differentiation using CRISPR activation by dCas9-VP64 (*Figure 5A* and *Figure 5—figure supplement 1*). dCas9-VP64 is a mutated Cas9 deficient in nuclease activity, which is fused to the VP64 effector domain and confers gene activation. Targeting of dCas9-VP64 to the *lncIrf8* promoter was achieved with specific gRNAs (*Figure 5—figure supplement 1I*). We also included targeting dCas9-VP64 to the *Irf8* promoter to study the interplay with the *lncIrf8* promoter. FACS sorted HoxB8 MPP expressing dCas9-VP64 and gRNA were subjected to DC differentiation and analyzed for *lncIrf8* and *Irf8* expression and DC subset composition (*Figure 5B–E* and *Figure 5—figure supplement 1A–E*).

Activation of the *lncIrf8* promoter by dCas9-VP64 caused a massive increase of *lncIrf8* expression at DC differentiation day 0 (*Figure 5C*). The activation of the *lncIrf8* promoter led also to *Irf8* upregulations at DC differentiation day 5, 7, and 9 compared to non-targeting control (*Figure 5A and C*). This demonstrates the positive impact of the *lncIrf8* promoter element on *Irf8* expression during DC differentiation and is in accord with the physical interaction of *lncIrf8* and *Irf8* promoters by Capture-C (*Figure 1B* and *Figure 1—figure supplement 1*).

Intriguingly, in *lncIrf8*-VP64 cells *lncIrf8* expression was downregulated at DC differentiation day 5, 7 and 9, while *Irf8* expression was upregulated (*Figure 5A and C*). This indicates a repressive effect of IRF8 on *lncIrf8* promoter and is in accord with IRF8 binding to the *lncIrf8* region by ChIP-seq (*Figure 1* and *Figure 1—figure supplement 1*). This observation suggests a negative feedback loop of IRF8 on the *lncIrf8* promoter during DC differentiation (*Figure 5A and C*).

Activation of *Irf8* promoter by dCas9-VP64 increased *Irf8* expression at DC differentiation day 0, 5, 7, and 9, while expression of *lncIrf8* was unaffected compared to non-targeting control (*Figure 5A and C*). As expected *Irf8* promoter activation led to higher pDC frequencies (*Figure 5B and D*), and also increased cDC1 frequencies compared to the non-targeting controls (*Figure 5B and E*). Importantly, *lncIrf8* promoter activation by dCas9-VP64 also increased cDC1 frequencies and this was particular prominent at day 9 of DC differentiation (*Figure 5B and E*). Frequencies of cDC2 were decreased at day 9 and other populations, including pDC, remained unchanged (*Figure 5B and D*, *Figure 5—figure supplement 1C–E*).

Taken together our CRISPR activation of the *lncIrf8* and *Irf8* promoters by dCas9-VP64 suggest a negative feedback loop of *Irf8* for pDC and cDC1 development.

## Negative feedback regulation of *lncIrf8* and *Irf8* promoters controls DC differentiation

To directly investigate the negative feedback loop of *Irf8* regulation, we repressed *lncIrf8* and *Irf8* promoters by targeted repression with dCas9-KRAB and analyzed the DC subsets during DC differentiation (*Figure 6A–F* and *Figure 5—figure supplement 1A, B, F-H*). dCas9-KRAB is a nuclease deficient Cas9 fused to the KRAB effector domain, which confers gene repression when positioned with specific gRNA (*Figure 5—figure supplement 1I*).

Targeted repression of the *Irf8* promoter decreased *Irf8* expression as expected but massively increased *lncIrf8* expression compared to non-targeting control (*Figure 6A and C*). This is very much in line with *Irf8* impacting *lncIrf8* expression by a negative feedback loop. Positioning the dCas9-KRAB

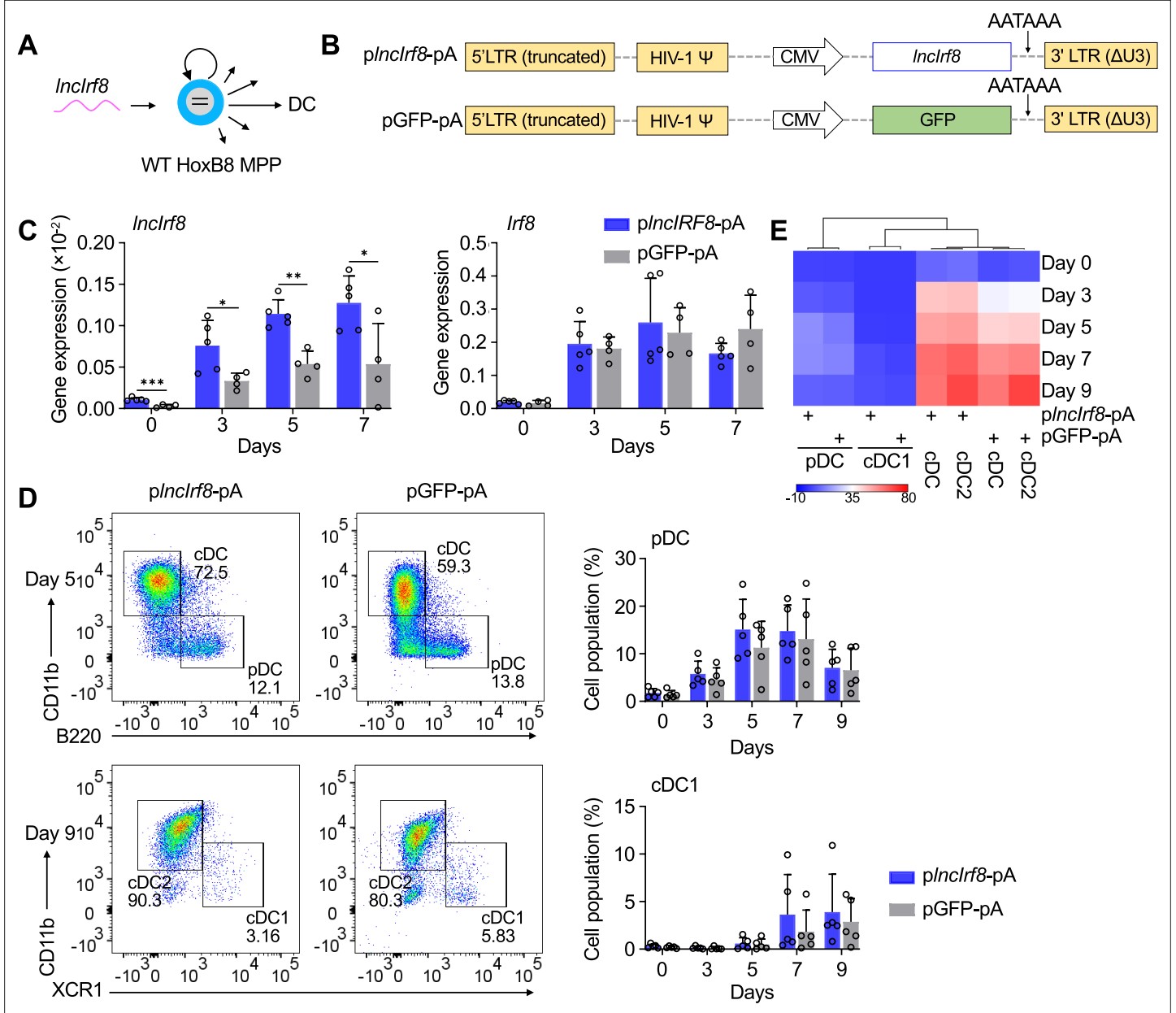

**Figure 4.** *lncIrf8* overexpression leaves pDC and cDC1 development unaffected. (**A and B**) Schematic representation of *lncIrf8* overexpression in WT HoxB8 MPP and of p*lncIrf8*-pA (*lncIrf8* overexpression) and pGFP-pA (control) plasmids. A polyA signal AATAAA for transcription termination was inserted at the 3' end of *lncIrf8* and GFP. (**C**) Gene expression of *lncIrf8* and *Irf8* in p*lncIrf8*-pA and pGFP-pA HoxB8 MPP on day 0, 3, 5, and 7 of Flt3L directed DC differentiation (n=4–5). Gene expression was by RT-qPCR and normalized to *GAPDH*. (**D**) Representative flow cytometry of DC subsets at day 5 and 9 of Flt3L directed DC differentiation of p*lncIrf8*-pA and pGFP-pA HoxB8 MPP. Quantification of pDC and cDC1 of living single cells on Flt3L directed DC differentiation day 0, 3, 5, 7, and 9 (n=5) is shown. Gating for pDC and cDC1 was as in ***Figure 2—figure supplement 1E***. (**E**) Heatmap representation of DC subsets of panel (**D**) at day 0, 3, 5, 7, and 9 of DC differentiation. Red, high frequency; white, intermediate frequency and blue, low frequency. Data represent mean ± SD of at least three independent experiments with different HoxB8 MPP clones of p*lncIrf8*-pA and pGFP-pA. *p<0.05, **p<0.01, ***p<0.001, multiple t-tests. Data that have no difference (p>0.05) are not labeled.

The online version of this article includes the following figure supplement(s) for figure 4:

**Figure supplement 1.** *lncIrf8* overexpression and rescue left DC differentiation unaffected.

repressor in the lnc*IRF8* promoter led to downregulation of both *Irf8* and *lncIrf8* expression, which confirms the *lncIrf8* promoter element acting on the *Irf8* promoter (***Figure 6A and C***).

Interestingly, targeted repression of the *Irf8* promoter severely compromised development of all DC subsets, including pDC, cDC1, and cDC2, compared to non-targeting controls and yielded CD11c⁺

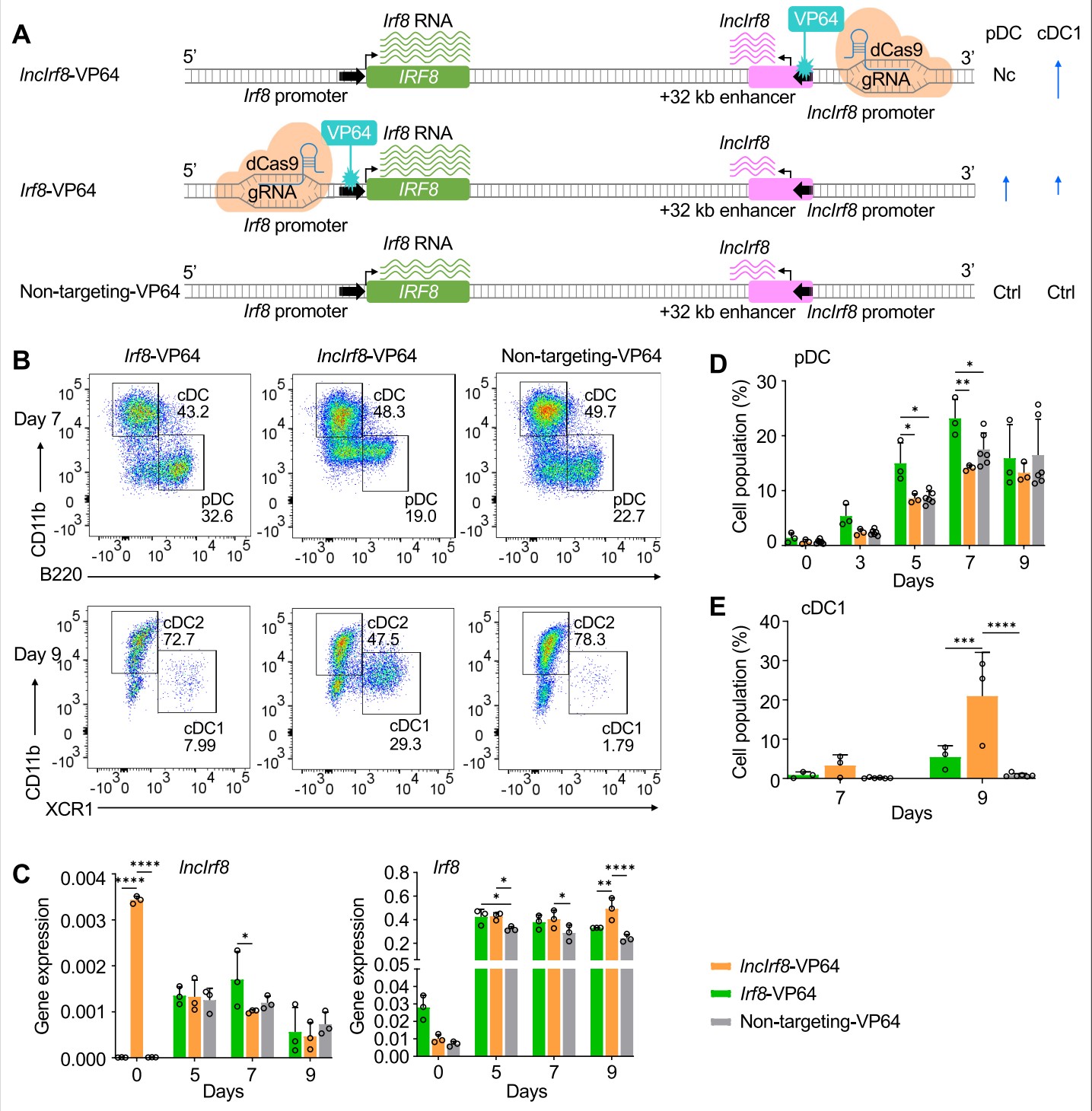

**Figure 5.** Activation of *lncIrf8* promoter promotes cDC1 development. (**A**) Schematic representation of *lncIrf8* and *Irf8* promoter activation (top and middle, respectively) by CRISPR activation with dCas9-VP64. gRNAs were positioned upstream of *lncIrf8* and *Irf8* TSS for gene activation. dCas9-VP64 cells with non-targeting gRNA were used as control (bottom). Green and purple wavy lines represent *Irf8* and *lncIrf8* RNA, respectively. The number of wavy lines indicates levels of gene transcription determined by RT-qPCR in (**C**). Different length of blue arrows represents the frequencies of pDC and cDC1 according to panel B, D and E. Nc, No change; Ctrl, Control. (**B**) Representative flow cytometry analysis of CRISPR activation targeting the *lncIrf8* and *Irf8* promoters at day 7 and 9 of Flt3L directed DC differentiation. Two non-targeting gRNAs were used as controls and one representative non-targeting gRNA is shown (Non-targeting-VP64). Top row, CD11b$^+$ B220$^-$ cDC and CD11b$^-$ B220$^+$ pDC at day 7 of Flt3L directed DC differentiation; bottom row, CD11b$^{low/-}$ XCR1$^+$ cDC1 and CD11b$^+$ XCR1$^-$ cDC2 at day 9 of DC differentiation. For gating strategy see ***Figure 5—figure supplement***

*Figure 5 continued on next page*

*Figure 5 continued*

*1B*. (**C**) Gene expression of *lncIrf8* and *Irf8* in *lncIrf8*-VP64, *Irf8*-VP64 and non-targeting-VP64 HoxB8 MPP on day 0, 5, 7, and 9 of Flt3L directed DC differentiation (n=3). Gene expression analysis was by RT-qPCR and data are normalized to *GAPDH*. Data represent mean ± SD of three independent experiments. *p<0.05, **p<0.01, ****p<0.0001, two-way ANOVA, Tukey's multiple comparisons test. Data that have no difference (p>0.05) are not labeled. (**D and E**) Quantification of pDC and cDC1 in percent of living single cells as in panel (**B**) on various days of Flt3L directed DC differentiation (n=3). Non-targeting-VP64 refers to both non-targeting-VP64 controls (n=6). Data represent mean ± SD of 3 independent experiments. *p<0.05, **p<0.01, ***p<0.001, ****p<0.0001, two-way ANOVA, Tukey's multiple comparisons test. Data that have no difference (p>0.05) are not labeled.

The online version of this article includes the following figure supplement(s) for figure 5:

**Figure supplement 1.** CRISPR/dCas9 mediated gene activation and repression, and gRNA positions.

cells with progenitor-like spherical morphology (***Figure 6B and D–F***, ***Figure 5—figure supplement 1F–H***). This reemphasizes the pivotal role of IRF8 for DC development known from studies on *Irf8* knockout mice (***Murakami et al., 2021***; ***Schiavoni et al., 2002***; ***Sichien et al., 2016***; ***Tsujimura et al., 2003***). In addition, targeted repression of the *lncIrf8* promoter (*lncIrf8*-KRAB) also compromised pDC and cDC1 development (***Figure 6B, D and E***). This result is very similar to the *lncIrf8* promoter KO analyzed above (***Figure 2A–C***).

All these findings support a model of *Irf8* regulating its own expression by a negative feedback loop acting on the *Irf8* +32 kb enhancer to limit *Irf8* autoactivation (***Figure 7***). This regulatory *Irf8* +32 kb enhancer element is marked by *lncIrf8*. *Irf8* expression starts in CDP and further increases in pDC and cDC1, with particularly high expression in pDC (***Figure 1A***, ***Figure 7A and B***, ***Figure 1—figure supplement 1***). The increase in *Irf8* expression is proposed to be due to an increase in the interactions of the *Irf8* promoter with upstream and downstream sequences. In pDC, the *Irf8* promoter-enhancer interactions are more with upstream chromatin regions (***Figure 1C*** and ***Figure 7A***), which relates to high *Irf8* expression. In cDC1, the *Irf8* promoter interactions are stronger with the regions downstream of *Irf8* (***Figure 1C*** and ***Figure 7A***) and the *Irf8* +32 kb enhancer marked by *lncIrf8* confers transcriptional repression.

In our model, we propose an IRF8 repressor complex that differentially acts on the *Irf8* +32 kb enhancer element in a DC subset specific manner to limit *Irf8* autoactivation. In cDC1 the +32 kb enhancer is repressed by the IRF8 repressor complex through negative feedback inhibition (prominent IRF8 binding in cDC by ChIP-seq; ***Figure 7A and B***), which limits *Irf8* autoactivation and expression. Conversely, in pDC there is less IRF8 repressor complex binding to the +32 kb enhancer, which results in high *Irf8* and *lncIrf8* transcription (***Figure 7A and B***). Recapitulating the IRF8 repressor complex with dCas9-KRAB and targeting *lncIrf8* promoter in the +32 kb enhancer reduced *Irf8* expression (***Figure 7C***). Conversely, activation of the +32 kb enhancer boosted *lncIrf8* and *Irf8* expression (***Figure 7C***). Thus, an intricate feedback loop of IRF8 on the +32 kb enhancer orchestrates *Irf8* expression and thus DC differentiation.

## Discussion

Hematopoiesis is a particularly well studied stem cell system and therefore provides an excellent model for studying TF in lineage commitment and cell differentiation, and the molecular principles involved (***Belz and Nutt, 2012***; ***Graf and Enver, 2009***; ***Laurenti and Göttgens, 2018***; ***Nutt and Chopin, 2020***). This work revealed a previously unrecognized negative feedback loop of *Irf8* in DC differentiation and shows how *Irf8* autoactivation is controlled and ultimately limited. IRF8 is crucial for DC lineage specification both in humans and mice (***Anderson et al., 2021***; ***Belz and Nutt, 2012***; ***Cabeza-Cabrerizo et al., 2021***; ***Cytlak et al., 2020***; ***Kurotaki et al., 2019***; ***Nutt and Chopin, 2020***). *Irf8* is upregulated by autoactivation via the +32 kb enhancer (***Grajales-Reyes et al., 2015***; ***Lin et al., 2015***). However, how *Irf8* expression is controlled at late stages of DC differentiation and eventually limited is not known. Here we demonstrate that *Irf8* expression is limited by a negative feedback loop via a sequence element marked by *lncIrf8* in the *Irf8* +32 kb enhancer.

*Irf8* expression in hematopoiesis is regulated by its flanking enhancers, which determine lineage specification and DC subset development (***Bagadia et al., 2019***; ***Durai et al., 2019***; ***Grajales-Reyes et al., 2015***; ***Murakami et al., 2021***; ***Schönheit et al., 2013***). Frequently, enhancers are identified by ATAC-seq and ChIP-seq (***Durai et al., 2019***; ***Murakami et al., 2021***), and here we embarked on a different approach and searched for eRNA and enhancer-associated lncRNA by RNA-seq. We

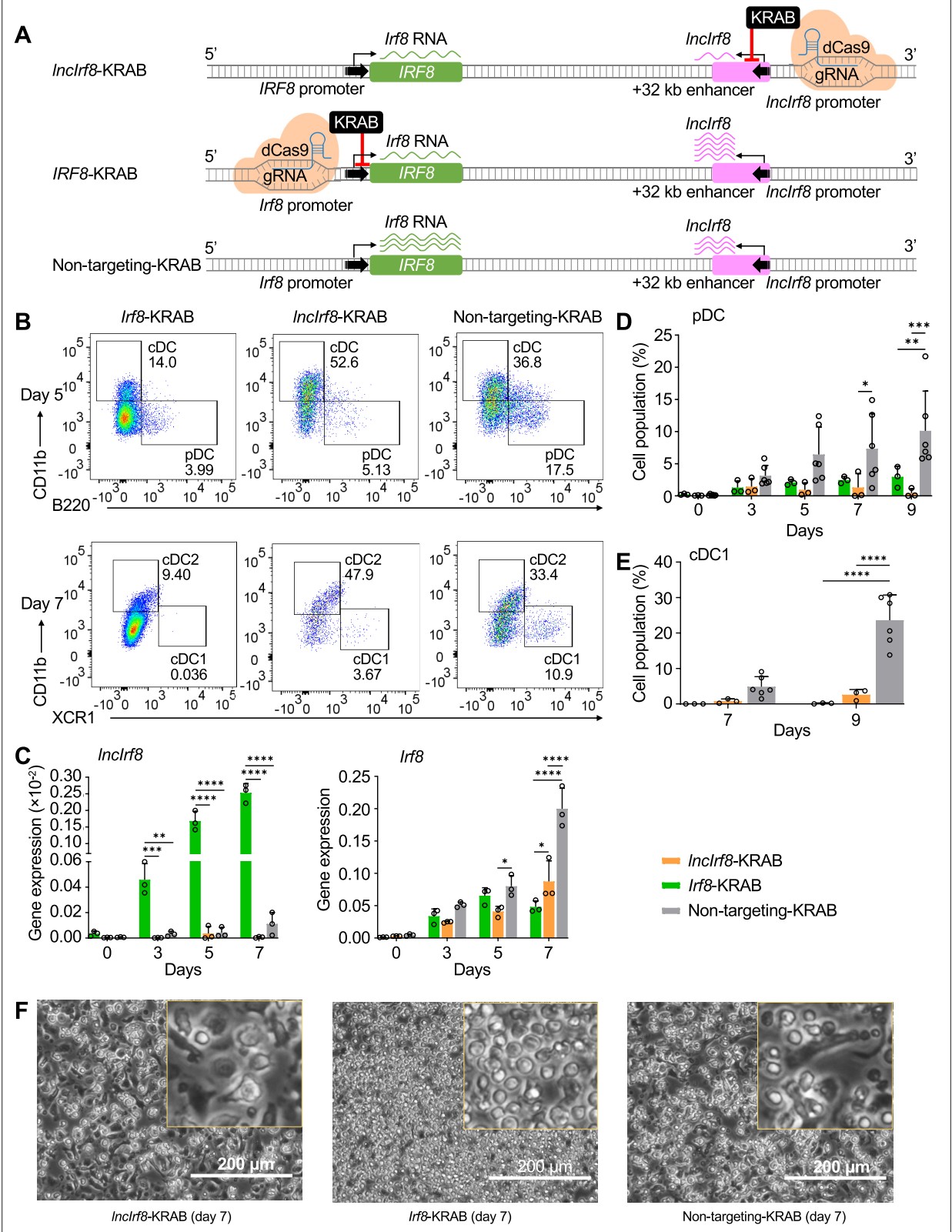

**Figure 6.** Repression of *lncIrf8* promoter compromises pDC and cDC1 development. (**A**) Schematic representation of *lncIrf8* and *Irf8* promoter repression (top and middle, respectively) by CRISPR interference with dCas9-KRAB. gRNAs were positioned downstream of *lncIrf8* and *Irf8* TSS to block gene transcription. dCas9-KRAB cells with non-targeting gRNA were used as control (bottom). Green and purple wavy lines represent *Irf8* and *lncIrf8* RNA, respectively. The number of wavy lines indicates levels of gene transcription determined by RT-qPCR in (**C**). (**B**) Representative flow cytometry

*Figure 6 continued on next page*

*Figure 6 continued*

analysis of CRISPR interference targeting the *lncIrf8* and *Irf8* promoters at day 5 and 7 of Flt3L directed DC differentiation. Two non-targeting gRNAs were used as controls and one representative non-targeting gRNA is shown (Non-targeting-KRAB). cDC and pDC at day 5 and cDC1 and cDC2 at day 7 of Flt3L directed DC differentiation are shown similar to *Figure 5B*. (**C**) Gene expression of *lncIrf8* and *Irf8* in *lncIrf8*-KRAB, *Irf8*-KRAB and non-targeting-KRAB HoxB8 MPP on day 0, 3, 5, and 7 of Flt3L directed DC differentiation (n=3). Gene expression analysis was by RT-qPCR and data are normalized to *GAPDH*. Data represent mean ± SD of three independent experiments. *p<0.05, **p<0.01, ***p<0.001, ****p<0.0001, two-way ANOVA, Tukey's multiple comparisons test. Data that have no difference (p>0.05) are not labeled. (**D and E**) Quantification of pDC and cDC1 in percent of living single cells as in panel (**B**) on various days of Flt3L directed DC differentiation (n=3). Non-targeting-KRAB refers to both non-targeting-KRAB controls (n=6). Data represent mean ± SD of three independent experiments. *p<0.05, **p<0.01, ***p<0.001, ****p<0.0001, two-way ANOVA, Tukey's multiple comparisons test. Data that have no difference (p>0.05) are not labeled. (**F**) Representative phase-contrast microscopy image of *lncIrf8*-KRAB, *Irf8*-KRAB and non-targeting-KRAB on day 7 of Flt3L directed DC differentiation. Scale bar: 200 µm.

identified a novel pDC-specific lncRNA (*lncIrf8*) transcribed from the *Irf8* +32 kb enhancer. *lncIrf8* itself lacks activities in DC differentiation but a *lncIrf8* promoter element is crucial for pDC and cDC1 development. Upon deletion of this *lncIrf8* promoter element pDC and cDC1 development was severely compromised, demonstrating that this sequence is important for both pDC and cDC1 development.

We propose a model where *Irf8* expression during DC differentiation is in a first step initiated and activated through flanking enhancers, including the +32 kb enhancer by autoactivation (*Grajales-Reyes et al., 2015*; *Lin et al., 2015*; *Figure 7*). In a second step, the *lncIrf8* promoter element confers feedback inhibition, which limits *Irf8* expression. This feedback inhibition is different for pDC and cDC1, both of which express high levels of *Irf8* (*Bornstein et al., 2014*; *Grajales-Reyes et al., 2015*; *Lin et al., 2015*). In cDC1, *Irf8* expression is attributed to *Irf8* autoactivation through the +32 kb enhancer driven by the BATF3-IRF8 complex (*Durai et al., 2019*; *Grajales-Reyes et al., 2015*). In pDC, *Irf8* expression is even higher than in cDC1 (*Bornstein et al., 2014*; *Lin et al., 2015*), which we propose is due to less feedback inhibition at late stages of DC differentiation.

A candidate for mediating *Irf8* feedback inhibition is IRF8 itself, since IRF8 works as transcriptional activator or repressor, depending on context. IRF8 activator or repressor function depends largely on the heterodimers (or even heterotrimers) with partner TF that bind to specific DNA sequences (*Chang et al., 2018*; *Huang et al., 2016*; *Humblin et al., 2017*; *Tamura et al., 2015*; *Yoon et al., 2014*). Modifications on IRF8 protein, such as phosphorylation and small molecule conjugation, also alter IRF8 activity (*Chang et al., 2012*; *Konieczna et al., 2008*). Potential IRF8 heterodimer partners, to form repressor complexes, are ETV6 or IRF2 (*Huang et al., 2016*; *Humblin et al., 2017*; *Lau et al., 2018*). The IRF8 repressor complex is proposed to bind to the +32 kb enhancer in cDC but not in pDC. This notion is in line with a prominent IRF8 signal at the +32 kb enhancer in cDC, which is absent in pDC (*Durai et al., 2019*; *Grajales-Reyes et al., 2015*; *Figure 1A*, *Figure 7A and B*, *Figure 1—figure supplement 1* and *Figure 2—figure supplement 2A*).

Further support of our model stems from our CRIPSR activation/interference experiments (*Figure 7C*). CRIPSR activation of *Irf8* promoter by dCas9-VP64 mimics *Irf8* up-regulation during DC differentiation and causes an increase in pDC and cDC1 (*Figure 5*). CRIPSR interference of *lncIrf8* promoter by dCas9-KRAB recapitulates transcriptional repressor binding to the +32 kb enhancer and causes *Irf8* promoter inhibition (*Figure 6*).

We extended our study to delineate the chromatin conformation of the *Irf8* promoter with flanking sequences by Capture-C. The *Irf8* promoter was found to interact with its flanking enhancers already at the CDP stage and then interactions with specific upstream and downstream sequences are proposed to guide pDC and cDC specification, respectively. This is in accord with previous studies where some of these *Irf8* flanking enhancers were required to maintain high levels of *Irf8* expression (*Anderson et al., 2021*; *Murakami et al., 2021*).

We demonstrate that deletion of the *lncIrf8* promoter element severely decreased *Irf8* expression and abolished both pDC and cDC1 development in vitro and in vivo upon cell transplantation. These results are very similar to a previous study on cDC1 specific +32 kb enhancer knockout mice, which demonstrates the impact of +32 kb enhancer for cDC1 development in vivo (*Durai et al., 2019*). The *lncIrf8* promoter is located in close proximity to the cDC1 specific +32 kb enhancer and thus can be expected to have overlapping functions. Indeed, deletion of the cDC1 +32 kb enhancer in HoxB8 MPP showed some similar activities on DC differentiation as the deletion of the *lncIrf8* promoter element, including regulation of *lncIrf8* and *Irf8* expression, but mainly affected cDC1 differentiation. These observations indicate that the *lncIrf8* promoter element has further functions compared to the

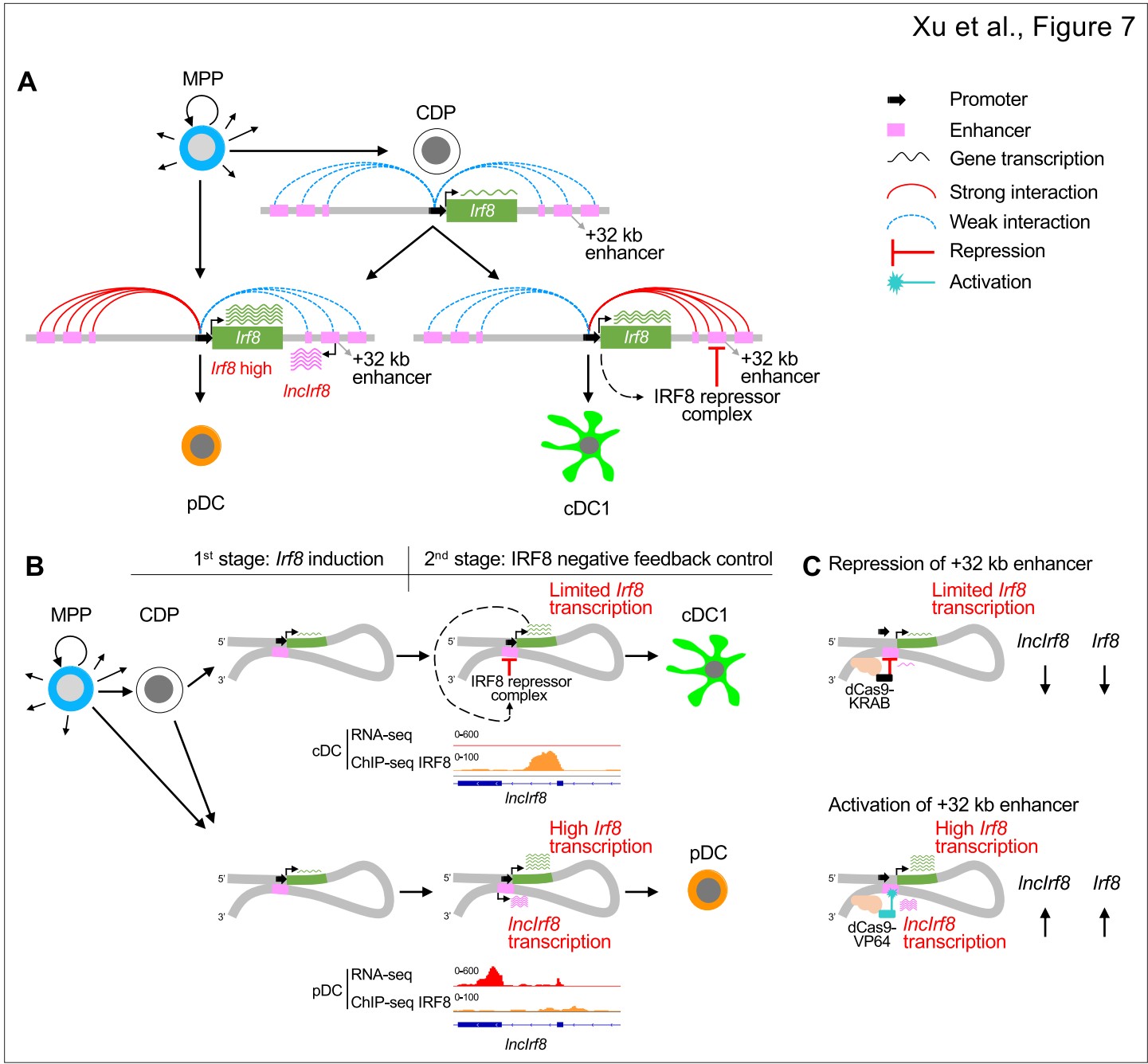

**Figure 7.** Negative feedback loop of *Irf8* through +32 kb enhancer governs DC differentiation. (**A**) Schematic representation of *Irf8* gene regulation during DC differentiation. *Irf8* transcription is induced at the CDP stage by its flanking enhancers, including the +32 kb enhancer, and is further increased in pDC and cDC1 (green wavy lines; the number of wavy lines indicates levels of gene transcription as in *Figure 5A* and *Figure 6A*). The increase in *Irf8* expression is due to an increase in the interactions of the *Irf8* promoter with upstream and downstream sequences. *Irf8* promoter interactions with upstream sequences are stronger in pDC and *Irf8* promoter interactions with downstream sequences are stronger in cDC1. In pDC the +32 kb enhancer marked by *lncIrf8* is less repressed by IRF8 repressor complex compared to cDC1, resulting in particularly high *Irf8* expression and *lncIrf8* transcription in pDC (purple wavy lines). This negative feedback inhibition of IRF8 on the +32 kb enhancer allows *Irf8* to regulate its own expression and thus DC differentiation. (**B**) Negative feedback regulation of *Irf8* through the +32 kb enhancer in cDC1 and pDC specification as described in panel (**A**). The RNA-seq and IRF8 ChIP-seq data are shown as in *Figure 1A*, *Figure 1—figure supplement 1* and *Figure 2—figure supplement 2A*. (**C**) Recapitulating *Irf8* and *lncIrf8* transcription by repression and activation of the *Irf8* +32 kb enhancer in HoxB8 MPP by targeted dCas9-KRAB and dCas9-VP64, respectively. Green and purple wavy lines represent gene expression as described in *Figure 5A* and *Figure 6A*.

cDC1 specific +32 kb enhancer, for example for pDC development, and the underlying mechanisms warrant further investigation.

eRNA and enhancer-associated lncRNA are indicative of enhancer activity, however whether the process of their transcription, the transcripts themselves, or both are functionally linked to enhancer activity, remains unclear (*Sartorelli and Lauberth, 2020*; *Statello et al., 2021*). Previous studies revealed that some eRNA and enhancer-associated lncRNA are indeed functionally connected with expression of the respective target genes (*Sartorelli and Lauberth, 2020*; *Statello et al., 2021*). Here, we found no apparent activity of *lncIrf8* on its own in pDC and cDC1 development, as demonstrated by *lncIrf8* overexpression and rescue experiments. *lncIrf8* appears to rather serve as an indicator for *Irf8* +32 kb enhancer activity. However, *lncIrf8* might have additional functions in DC biology, which are not revealed in the current study and remain to be identified.

In conclusion, by analyzing the gene expression and epigenetic profiles of the *Irf8* locus, we identified an enhancer element marked by *lncIrf8* that regulates *Irf8* and controls DC differentiation through a negative feedback loop. Our results suggest that *Irf8* regulates itself by its flanking enhancers in DC fate determination: First, *Irf8* induces its expression by autoactivation via flanking enhancers, including the *Irf8* +32 kb enhancer, to initiate DC differentiation, and second limits its expression at late stages via the *lncIrf8* promoter element within the same +32 kb enhancer by feedback inhibition. This molecular principle of feedback inhibition is expected to also apply to other TF and cell differentiation systems.

## Materials and methods
### Mice
Wild type C57BL/6, Mx-Cas9-GFP knock-in mice (*Kühn et al., 1995*; *Platt et al., 2014*; *Xu et al., 2022*), and CD45.1 recipient C57BL/6 mice were used in this study. Mice were kept under specific pathogen-free conditions in the central animal facility of RWTH Aachen University Hospital, Aachen, Germany. All the animal experiments were approved by the local authorities of the German State North Rhine-Westphalia, Germany according to the German animal protection law (reference number: 81–02.04.2018 .A228).

### Cell culture
Multipotent progenitors (MPP) were obtained from mouse bone marrow and expanded in vitro with a two-step protocol as described in *Felker et al., 2010*. Conditionally immortalized HoxB8 MPP were generated by retrovirus infection of bone marrow cells from wild-type or Mx-Cas9-GFP knock-in mice with an estrogen (E2) inducible HoxB8 estrogen receptor (ER) fusion gene (HoxB8-ER) (*Redecke et al., 2013*; *Xu et al., 2022*). MPP were grown in RPMI 1640 medium with 10% FCS (Gibco, 10270106), 100 U/ml penicillin/streptomycin, 2 mM L-glutamine and 50 µM β-ME with a four-cytokine cocktail of SCF, Flt3 ligand (Flt3L), IGF-1, and IL-6/soluble IL-6 receptor fusion protein (hyper-IL-6) as before (referred to as MPP growth medium) (*Felker et al., 2010*; *Lutz et al., 2022*; *Xu et al., 2022*) (Appendix 1-key resources table). E2 (1 µM) was added to activate HoxB8-ER and to maintain the conditionally immortalized state of HoxB8 MPP. Cell density was adjusted to 1.5 million cells/ml every day. HEK293T cells for retrovirus and lentivirus production were grown in DMEM supplemented with 10% FCS (PAA, A01125-499), 100 U/ml penicillin/streptomycin, 2 mM L-glutamine (Appendix 1-key resources table).

### In vitro DC differentiation with HoxB8 MPP
HoxB8 MPP were differentiated into DC using a two-step protocol modified from *Felker et al., 2010* and described in *Lutz et al., 2022*; *Xu et al., 2022*. In brief, 0.75 million cells/ml were grown in MPP growth medium with 50 ng/ml Flt3L (Peprotech, 300–19) and reduced E2 (0.01 µM) for two days and cell density was kept to 0.75 million cells/ml. To induce DC differentiation, HoxB8 MPP were then washed with PBS to remove cytokines and E2, and cultured in RPMI 1640 medium supplemented with FCS, penicillin/streptomycin, L-glutamine, β-ME (same concentrations as above), and Flt3L (50 ng/ml, Peprotech) (referred to as DC differentiation day 0). Partial medium changes were performed on differentiation day 3 and 6. Spontaneous DC differentiation of HoxB8 MPP was achieved simply by removing E2 from growth medium (SCF, Flt3L, IGF1 and hyper-IL6), and culturing the cells at 1.5 million cells/ml (*Lutz et al., 2022*; *Xu et al., 2022*).

## Nuclear-Titrated (NuTi) Capture-C

Wild-type bone marrow cell-derived MPP, CDP, cDC1, cDC2, and pDC were generated in vitro with the two-step protocol as described in *Felker et al., 2010*. Cell populations were sorted by BD FACSAria Ilu or BD FACSMelody, MPP are Gr1$^-$ CD117$^{hi}$ CD135$^{low/-}$, CDP are Gr1$^-$ CD117$^{int}$ CD135 $^+$ CD115$^+$, cDC1 are CD11c$^+$ CD11 b$^{low/-}$ XCR1$^+$, cDC2 are CD11c$^+$ CD11b$^+$ XCR1$^-$, pDC are CD11c$^+$ CD11b$^-$ B220$^+$.

The chromatin conformation capture (3 C) library preparation protocol used in this study was modified from *Downes et al., 2021*; *Downes et al., 2022* with the reagents listed in Appendix 1-key resources table. MPP, CDP, cDC1, cDC2 and pDC were fixed with formaldehyde (final concentration 2%) and subjected to nuclear isolation according to the protocol in *Downes et al., 2022*; *Li et al., 2019*. Nuclei (15–20 million per sample) were digested with DpnII and DNA fragments were ligated by T4 DNA HC ligase. DNA was extracted and purified with Phenol-Chloroform-Isoamyl alcohol (PCI, 25:24:1). DpnII digestion efficiency was determined by SYBR qPCR with the primers listed in Appendix 1-key resources table and the quality of 3 C libraries was investigated by agarose gel (1%) electrophoresis; 3 C samples were used only if the DpnII digestion efficiency was more than 70%.

For *Irf8* promoter viewpoint 2 oligonucleotide probes targeting *Irf8* promoter were designed with the design tool Oligo (https://oligo.readthedocs.io/en/latest/index.html). Oligonucleotide probes are positioned adjacent to the DpnII cut sites on a restriction fragment spanning the *Irf8* promoter (chr8:123,259,948–123,260,530) and 70 bp ssDNA biotinylated oligonucleotides were synthetized by Sigma-Aldrich (listed in Appendix 1-key resources table).

To enrich for fragments containing ligation events with *Irf8* promoter viewpoint, NuTi Capture-C was performed as previously described (*Downes et al., 2021*; *Downes et al., 2022*). Briefly, the 3 C libraries prepared from MPP, CDP, cDC1, cDC2 and pDC were sonicated using Covaris S220 to an average size of ~200 bp using standard settings recommended by the manufacturer (time: 180 s, duty cycle: 10%, peak incident power: 175 W, cycles per burst: 200, temperature: 5–9°C). End repair was performed with the NEBNext Ultra II kit (New England Biolabs, E7645S) using 2 µg of sonicated 3 C library in duplicate for each sample. Illumina NEBNext Indices (New England Biolabs, E7500S, and E7335S) were added with a total of 6 cycles of amplification to allow for multiplexing. After this step, duplicates were pooled to increase sample complexity. We enriched samples as per NuTi Capture-C protocol, with two capture rounds in a multiplexed reaction, using 2 µg of each indexed sample as an input for the first capture. The hybridization with biotinylated probes was performed with the KAPA Hyper Capture Reagent Kit (Roche, 9075828001). Each ssDNA biotinylated probe was used at a concentration of 2.9 nM, with a final pool concentration of 5.8 nM, and 4.5 µl of the pooled oligonucleotides were used per sample. Captured DNA was pulled-down with M-270 Streptavidin Dynabeads (Invitrogen, 65305), washed and amplified off the beads with a total of PCR 14 cycles. The DNA obtained after the first capture was used as an input in the second capture round. The experiments were performed for the first and the second biological replicate separately, and then sequenced with NextSeq 550 Illumina System with 300 paired-end or 150 paired-end, respectively.

## NuTi Capture-C data analysis

The Capture-C data was analyzed with CapCruncher (v0.1.1a1) pipeline (https://github.com/sims-lab/CapCruncher; *sims-lab, 2022*) in capture mode (*Downes et al., 2022*). The reads were aligned to the mm9 genome assembly with Bowtie2 (*Langmead and Salzberg, 2012*), with specific options: -p6--very-sensitive. Viewpoint coordinates used were: chr8:123,259,948–123,260,530, 1000 bp around viewpoint was excluded. The data were normalized to ~300 kb region around the viewpoint (chr8:123,132,865–123,433,117).

The Capture-C profiles in *Figure 1* and *Figure 1—figure supplement 1* show the mean number of unique interactions in two biological replicates, normalized to ~300 kb region around the viewpoint. Differential tracks were created by subtraction of the mean normalized tracks.

## ATAC-seq, ChIP-seq, RNA-seq, and scRNA-seq

ATAC-seq analysis of MPP, CDP, cDC1, cDC2, and pDC was performed by Omni-ATAC-seq (*Corces et al., 2017*) with minor modifications as described in *Li et al., 2019*. RNA-seq analysis and ChIP-seq analysis was done as previously described (*Allhoff et al., 2016*; *Lin et al., 2015*).

To determin lncRNA expression in mouse ex-vivo pDC, scRNA-seq data of BM and splenic pDC were downloaded from GSE114313 (*Rodrigues et al., 2018*) and BAM files were converted back into FASTQ files using bamtofastq (https://support.10xgenomics.com/docs/bamtofastq). For visualization of lncRNA expression, we created a custom reference genome of mm9 by following the tutorial: available here. Cellranger pipeline was then processed to generate an expression count matrix of pDC. We next used scanpy (*Wolf et al., 2018*) to analyze the scRNA-seq data. We filtered cells based on the number of detected genes (<200 or>3500), and the proportion of mitochondrial genes (>10%). After data quality control, we retained 7044 cells for splenic pDC and 8158 cells for BM pDC, respectively. We then log-normalized the count matrix using a scaling factor of 10,000 and selected the top 3000 highly variable genes, which were used for dimensionality reduction based on principle component analysis. The cells were visualized using uniform manifold approximation and projection (UMAP) (*Becht et al., 2019*).

## Plasmids

psPAX2 (Addgene #12260) and pMD2.G (Addgene #12259) for lentiviral packaging and envelope expressing plasmids were kind gifts from Didier Trono, EPFL, Lausanne, Switzerland. The gRNA expressing vector pLKO5.sgRNA.EFS.tRFP (Addgene #57823) and pLKO5.sgRNA.EFS.tRFP657 (Addgene #57824) were kind gifts from Benjamin Ebert, Harvard Medical School, Boston, USA (*Heckl et al., 2014*).

For *lncIrf8* overexpression and rescue, *lncIrf8* cDNA was introduced into polyA containing lentivirus vector pGFP-pA to generate p*lncIrf8*-pA. pGFP-pA was constructed from pCIG3 (Addgene #78264; *Caviness et al., 2014*) by Gibson Assembly (New England Biolabs, E5510S; *Gibson et al., 2009*). In brief, the WPRE element was removed and a polyA signal 'AATAAA' was inserted at the 3' end of GFP to construct pGFP-pA. *lncIrf8* cDNA was sub-cloned into pGFP-pA using XhoI and SalI with the primers shown in Appendix 1-key resources table to obtain p*lncIrf8*-pA. CRISPR activation and repression of *lncIrf8* and *Irf8* promoters were achieved by dCAS9-VP64_GFP (Addgene #61422) (*Konermann et al., 2015*) and pTet-KRAB-dCas9-GFP (*Xu et al., 2022*), respectively.

## Lentivirus infection

gRNAs, *lncIrf8*, dCas9-VP64-GFP, and KRAB-dCas9-GFP were delivered into HoxB8 MPP by lentiviral infection. Briefly, lentiviral particles were produced by calcium phosphate transfection (*Graham and van der Eb, 1973*) of HEK293T cells with psPAX2, pMD2.G, and the lentiviral transfer vector. At 48 and 72 hours after HEK293T cell transfection, supernatant containing virus particles was collected and concentrated using chondroitin sulfate sodium salt (CSS) and polybrene precipitation (*Landazuri et al., 2007*; Appendix 1-key resources table). HoxB8 MPP were infected twice with concentrated virus and subjected to Ficoll (Pancoll) purification to remove precipitate and dead cells.

## Genetically modified HoxB8 MPP cell lines

### *lncIrf8* promoter and cDC1 specific +32 kb enhancer KO:

Mx-Cas9-GFP HoxB8 MPP were used to generate *lncIrf8* promoter and cDC1 specific +32 kb enhancer knockout (KO) HoxB8 MPP by CRISPR/Cas9. Briefly, pairs of gRNAs each for *lncIrf8* promoter KO and cDC1 specific +32 kb enhancer KO (Appendix 1-key resources table) were designed with the IDT online gRNA design tool. For cDC1 specific +32 kb enhancer KO we additionally used the same gRNA sequences from *Durai et al., 2019* listed in Appendix 1-key resources table. gRNAs were cloned into pLKO5.sgRNA.EFS.tRFP and pLKO5.sgRNA.EFS.tRFP657 vectors (*Heckl et al., 2014*) using BsmBI-v2 (Appendix 1-key resources table), respectively.

One 10 cm dish (Bio-One) with 1.8 million HEK293T cells (70%–80% confluency) in 10 ml DMEM plus supplements (see above) was used to produce gRNA expressing lentivirus particles. HEK293T cells were transfected with 10 µg gRNA vector, 7.5 µg psPAX2, and 2.5 µg pMD2.G per gRNA by calcium phosphate transfection and lentivirus particles were harvested 48 and 72 h after transfection.

The gRNA expressing lentiviral particles were used to infect 3 million Mx-Cas9-GFP HoxB8 MPP. Cas9 and GFP expression were induced by mIFNα (1000 IU/ml; Appendix 1-key resources table) for 4 h, followed by FACS sorting for cells expressing the two gRNAs and Cas9 (GFP[+] RFP[+] RFP657[+] cells). Single-cell clones were obtained by single-cell FACS sorting or limiting dilution.

**Table 1.** Off-target analysis.

| Potential off-targets | | | KO bulk (100 cells) | KO single-cell clones | | | | | Gene expression | | | |
|---|---|---|---|---|---|---|---|---|---|---|---|---|
| | | | | 6 | 7 | 19 | 21 | 24 | MPP | CDP | cDC | pDC |
| gRNA1 | Chr 13: −48032179 | Gm36101 | No | No | No | No | No | No | No | No | No | No |
| | Chr 2: −131293076 | Non-coding | No | No | No | No | No | No | No | No | No | No |
| | Chr 1: −136050990 | AsCl5 | 16 bp deletion | 16 bp deletion | 16 bp deletion | No | 1 bp insertion | No | No | No | No | No |
| | Chr 6:+119352090 | CACNA2D4 | No | No | No | No | No | No | No | No | No | Yes |
| gRNA2 | Chr 3:+33982811 | Non-coding | 3–29 bp deletion; 1–3 bp insertion | 3–29 bp deletion | 3 bp deletion | 1–2 bp insertion | No | No | No | No | No | No |
| | Chr 2: −132060887 | Non-coding | No | No | No | No | No | No | No | No | No | No |
| | Chr 2: −150624255 | Non-coding | No | No | No | No | No | No | No | No | No | No |

Top potential off-targets of gRNA1 and gRNA2 predicted by CRISPR-Cas9 gRNA checker (see Materials and methods) were analyzed in *lncIrf8* promoter KO bulk culture and KO single-cell clones by PCR analysis of genomic DNA. Potential off-target genes (coding) and non-coding sequences are listed. The absence of off-targets (No) and off-target deletions/insertion are shown.

Single-cell clones were genotyped by genomic PCR with primers listed in Appendix 1-key resources table. PCR products were sequenced by Eurofins Genomics and sequences were analyzed by Snap-Gene. Potential off-targets were routinely predicted by online software tools such as CRISPR-Cas9 gRNA checker (https://eu.idtdna.com/site/order/designtool/in-dex/CRISPR _SEQUENCE). For *lncIrf8* promoter KO, 5 out of 71 single-cell colonies with homozygous deletions were subjected to off-target analysis. The top 2–5 predicted coding or non-coding targets were selected and HoxB8 MPP clones without off-target effects, or where off-target effects occurred in genes that were not expressed in MPP, CDP and DC subsets, were used for further studies (*Table 1*). For cDC1 specific +32 kb enhancer KO, 5 out of 165 single-cell clones with homozygous deletions were subjected to DC differentiation and further analyzed. Potential off-target genes of self-designed gRNAs for cDC1 +32 kb enhancer KO were not expressed in MPP, CDP, and DC subsets, and thus these gRNAs were used in the study.

### *lncIrf8* overexpression and *lncIrf8* knockout rescue

*lncIrf8* overexpression was performed in wild-type HoxB8 MPP. Lentiviral particles expressing *lncIrf8* were produced from ten 6 cm dishes (Bio-One), each consisting of 0.75 million HEK293T cells (70–80% confluency) in 5 ml DMEM with supplements (see above). HEK293T cells were transfected with 5 µg p*lncIrf8*-pA or pGFP-pA, 2.5 µg psPAX2, and 2.5 µg pMD2.G. Lentivirus particles were concentrated as above and used to infect 0.5 million HoxB8 MPP; single-cell clones were generated by limiting dilution and genotyped using the primers listed in Appendix 1-key resources table. Two out of 47 HoxB8 MPP colonies with p*lncIrf8*-pA and 3 out of 19 HoxB8 MPP colonies with pGFP-pA (control) were expanded and subjected to Flt3L-directed DC differentiation.

*lncIrf8* knockout rescue was carried out in *lncIrf8* promoter KO HoxB8 MPP. FACS sorted cells that genotyped as *lncIrf8* promoter KO homozygous deletion cells, were infected with lentiviral particles expressing *lncIrf8*. Lentiviral infection conditions were the same as for *lncIrf8* overexpression in wild-type HoxB8 MPP (see above).

### CRISPR activation and CRISPR interference

CRISPR activation and CRISPR interference were performed by infecting wild-type HoxB8 MPP with lentiviral particles expressing dCAS9-VP64_GFP and pTet-KRAB-dCas9-GFP, respectively. The virus particles were produced as in the *lncIrf8* overexpression experiments. In brief, virus particles from ten 6 cm dishes were used to infect 0.5 million wild-type HoxB8 MPP, followed by FACS sorting for GFP+ cells expressing dCas9-VP64 or dCas9-KRAB. Doxycycline (1 µg/ml) was used to induce dCas9-KRAB expression 2 days before cell sorting.

gRNAs targeting *lncIrf8* and *Irf8* promoters were cloned into pLKO5.sgRNA.EFS.tRFP as above. The dCas9-VP64-GFP and dCas9-KRAB-GFP HoxB8 MPP were then infected with specific gRNAs for gene activation and repression. The conditions for gRNA infection were the same as in *lncIrf8*

promoter KO experiments. Doxycycline (1 µg/ml) was given to the sorted dCas9-KRAB-GFP cells every 3 days to ensure sustained dCas9-KRAB expression.

## Flow cytometry analysis and cell sorting

DC subsets were analyzed by flow cytometry using FACS Canto II or LSR Fortessa (both from BD Biosciences). The information for flow cytometry antibodies is shown in Appendix 1-key resources table. For live/dead staining, cells were incubated with 3 µl 7-AAD per test for 5–10 min before flow cytometry measurement. Cells were sorted by FACS Aria IIu or FACS Melody, and flow cytometry data were analyzed by FlowJo V10 (all from BD Biosciences). Data on DC frequencies were subjected to the hierarchical clustering and represented in heatmap format using the online tool MORPHEUS (https://software.broadinstitute.org/morpheus/).

## Cell transplantation

CD45.1 recipient mice were sublethal irradiated (6.0 Gy, CP-160 Faxitron) 1 day before transplantation. *lncIrf8* promoter KO and control HoxB8 MPP (single-cell clones) were expanded in vitro as described above. Cells were injected into recipient mice via the tail vein (5 million cells in 100 µl PBS per mouse). Donor cells from bone marrow and spleen were subjected to flow cytometry analysis at 7 and 14 days after cell transplantation.

## RT-qPCR

RNA was isolated by using the NucleoSpin RNA kit (Macherey-Nagel, 740955.250) according to the manufacturer's instructions. RNA was subjected to cDNA synthesis using the High-Capacity cDNA Reverse Transcription Kit (Applied Biosystems, 4368814) with Murine RNase Inhibitor (New England Biolabs, M0314S) (Appendix 1-key resources table). RT-qPCR was performed by a StepOnePlus Real-Time PCR system (Applied Biosystems) with SYBR-green fluorescence (Applied Biosystems, 4385610). The primers for qPCR were listed in Appendix 1-key resources table. Mouse *GAPDH* was used for normalization of *lncIrf8* and *Irf8* gene expression.

## Identification of *lncIrf8*

De-novo transcript assembly of RNA-seq data was used to search for unknown transcripts with no coding potential and this identified the pDC specific lncRNA *Tcons_00190250* (referred to as *lncIrf8*) and the cDC1-specific lncRNA *Tcons_00190258*. In brief, paired end 2×100 bps reads from RNA-seq of MPP, CDP, cDC1, cDC2, and pDC were aligned to mm9 genome using Star aligner (version 2.4) (*Dobin et al., 2013*) and run for Cufflinks (version 2.0; *Trapnell et al., 2012*). Data were subjected to lenient filtering with the parameters: min isoform fraction 0.1% and pre-RNA-fraction of 0.1%, and ribosomal genes were also filtered. Next, all the predicted transcripts were merged with cuffmerge and all transcripts with no overlap with known transcripts in mouse GENCODE were selected for further analysis (*Frankish et al., 2019*). The coding potential and conservation of coding elements of the lncRNAs were checked with CPAT (*Wang et al., 2013*) and PhyloCSF (*Lin et al., 2011*), respectively. *lncIrf8* (*Tcons_00190250*) and *Tcons_00190258* show low scores in both analyses, which faithfully supports their role as non-coding genes and exhibit low cross-species conservation.

To further characterize the major transcripts of *lncIrf8*, 3' and 5' end Rapid Amplification of cDNA Ends (RACE) PCR was performed using template-switching RT enzyme mix (New England Biolabs, M0466) and TA cloning kit (Thermo Fisher Scientific, K202020) (Appendix 1-key resources table). The primers (listed in Appendix 1-key resources table) used for RACE PCR were synthesized by Eurofins Genomics except for 5' RACE-TSO, which was from IDT.

## 3' RACE PCR

Reverse transcript (RT) and template-switching: 4 µl (10 ng to 1 µg) total RNA (from DC differentiation day 5) were incubated at 80°C for 3 min and cooled rapidly on ice. RNA was then incubated with template-switching RT buffer, 1 mM dNTP, 5 mM DTT, 10 µM QT primer (*Scotto-Lavino et al., 2006*) and 1 µl RT enzyme in 10 µl at room temperature for 5 min, followed by 1 hr at 42°C, 10 min at 50 °C, and 85 °C for 5 min to inactive the RT enzyme mix and sample was then kept at 4 °C and diluted with 20 µl MilliQ water.

First-round PCR: 1 µl of diluted sample was subjected to the first round PCR with Q0 primer and 3' RACE-GSP-*lncIrf8*-F1 (10 µM) using Q5 high fidelity DNA polymerase. Second-round PCR: the PCR products from the first round PCR (1:20 dilution) was then used as template for the second round of PCR by using Q1 primer (10 µM), 3' RACE-GSP-*lncIrf8*-F2 (10 µM). Q0 and Q1 primers and the PCR programs are described in *Scotto-Lavino et al., 2006*. Products from the second-round PCR were purified using the PCR clean-up kit (Macherey-Nagel, 740609.50).

A-tailing and TA cloning: a reaction containing 5 µl of PCR clean-up product in Taq buffer, 1 mM MgCl$_2$, 0.4 mM dATP and 1 µl Taq DNA polymerase in 25 µl was prepared and incubated at 70 °C for 20 min for A-tailing of PCR products. Five µl of the A-tailed products were subjected to TA cloning into pCR2.1 vector according to the manufacturer's instruction followed by Sanger sequencing.

## 5' RACE PCR

In order to identify the TSS of *lncIrf8*, template RNA from bone marrow cell-derived pDC was used to perform 5' RACE PCR using template-switching RT enzyme mix (New England Biolabs, M0466) according to the manufacturer's instructions. Template switching was by 5' RACE-GSP-*lncIrf8*-R2 primer (10 µM) and template switching oligo (TSO) (75 µM). Similar to the 3' RACE PCR, two rounds of PCR were used to improve PCR specificity. In brief, 5' RACE-GSP-*lncIrf8*-R2 (10 µM) and TSO-specific primer (10 µM) were used to perform the first-round PCR, 5' RACE-GSP-*lncIrf8*-R1 primer (10 µM) and the TSO-specific primer (10 µM) were used to perform the second round PCR, followed by fragments A-tailing, TA cloning, and Sanger sequencing as described above for 3' RACE PCR.

## Identification of transcription factor binding sites

Transcription factor binding sites (TFBS) in the *Irf8* +32 kb enhancer were predicted using a motif matching tool based on the MOODS (*Korhonen et al., 2009*) and position weight matrixes (PWMs) were obtained from the JASPAR database (*Fornes et al., 2020*). The bit-score cut-off thresholds were determined by applying the dynamic programming approach as described in *Wilczynski et al., 2009* with an FPR of 0.0001. DC TF were considered and are depicted.

## Statistical analysis

Statistical analyses were performed using Prism (GraphPad). Unpaired t test and Multiple t-tests were used to compare data from two groups, two-way ANOVA with Tukey's multiple comparisons test was used to analyze data from more than two groups.

## Acknowledgements

We acknowledge the support of the Interdisciplinary Center for Clinical Research Aachen (IZKF Aachen) FACS Core Facility and Genomics Facility. We thank Magdalena Karpinska and Marieke Oudelaar for help with the Capture-C experiments and analysis, Susanne Schmitz and Paul Wanek for assistance, Stefan Rose-John for hyper-IL-6, Lisa Weixler and Carmen Schalla for help with enzyme purification, and Thomas Hieronymus for valuable discussion and suggestions. Part of this work was supported in part by funds from German Research Foundation (DFG) to MZ and from the Germany Ministry of Science and Technology (BMBF - Fibromap) and the Interdisciplinary Center for Clinical Research Aachen (IZKF Aachen) from the RWTH Medical Faculty to IC. HX was supported by a fellowship of China Scholarship Council (CSC) (Grant number 202008080170). MAST was funded by CAPES-Alexander von Humboldt postdoctoral fellowship (99999.001703/2014–05) and donation by U Lehmann.

## Additional information

### Funding

| Funder | Grant reference number | Author |
| --- | --- | --- |
| German Research Foundation | | Martin Zenke |

| Funder | Grant reference number | Author |
|---|---|---|
| German Ministry of Science and Technology | Fibromap | Ivan G Costa |
| Interdisciplinary Center for Clinical Research Aachen | | Ivan G Costa Martin Zenke |
| China Scholarship Council | 202008080170 | Huaming Xu |
| CAPES-Alexander von Humboldt Foundation | 99999.001703/2014-05 | Marcelo AS de Toledo |

The funders had no role in study design, data collection and interpretation, or the decision to submit the work for publication.

## Author contributions

Huaming Xu, Conceptualization, Resources, Data curation, Formal analysis, Validation, Investigation, Visualization, Methodology, Writing – original draft, Writing – review and editing; Zhijian Li, Chao-Chung Kuo, Resources, Data curation, Software, Visualization, Methodology; Katrin Götz, Investigation, Methodology, Writing – review and editing; Thomas Look, Conceptualization, Investigation, Methodology; Marcelo AS de Toledo, Kristin Seré, Conceptualization, Investigation, Methodology, Writing – review and editing; Ivan G Costa, Conceptualization, Resources, Data curation, Software, Formal analysis, Supervision, Funding acquisition, Investigation, Methodology, Writing – review and editing; Martin Zenke, Conceptualization, Resources, Data curation, Formal analysis, Supervision, Funding acquisition, Methodology, Writing – original draft, Project administration, Writing – review and editing

## Author ORCIDs

Ivan G Costa (ID) http://orcid.org/0000-0003-2890-8697
Martin Zenke (ID) http://orcid.org/0000-0002-1107-3251

## Ethics

All the animal experiments were approved by the local authorities of the German Federal State North Rhine-Westphalia, Germany according to the German animal protection law (reference number 81-02.04.2018.A228).

## Decision letter and Author response

Decision letter https://doi.org/10.7554/eLife.83342.sa1
Author response https://doi.org/10.7554/eLife.83342.sa2

# Additional files

## Supplementary files

• MDAR checklist

## Data availability

ATAC-seq (MPP, CDP and cDC2), Capture-C targeting *Irf8* promoter (MPP, CDP, cDC1, cDC2 and pDC), IRF8 ChIP-seq (cDC and pDC), and RNA-seq (MPP, CDP, cDC1, cDC2 and pDC) data generated in this study have been deposited in Gene Expression Omnibus and are accessible through GEO Series accession numbers GSE198651.ATAC-seq data of cDC1 and pDC (GSE118221) are published (*Lin et al., 2015*), and the ATAC-seq data of MPP, CDP and cDC2 of the same study are described here (GSE198651). ChIP-seq of CTCF in DC (GSE36099) (*Garber et al., 2012*), H3K27ac (GSE73143) (*Allhoff et al., 2016*), H3K4me1 and PU.1 (GSE57563) (*Allhoff et al., 2014*), H3K4me3 and H3K9me3 (GSE64767) (*Lin et al., 2015*) in MPP, CDP, cDC and pDC were re-analyzed in this study. scRNA-seq data were reanalyzed from GSE114313 (*Rodrigues et al., 2018*). Bulk RNA-seq data of splenic pDC, cDC1 and cDC2 were reanalyzed from GSE188992 (*Pang et al., 2022*).The sequence of the pDC specific lncRNA (*lncIrf8* identified by RACE PCR) and the cDC1 specific lncRNA (*Tcons_00190258* identified by RNA-seq) has been submitted to GenBank. The GenBank accession numbers for *lncIrf8* and *Tcons_00190258* are ON134061 and ON134062, respectively.

The following datasets were generated:

| Author(s) | Year | Dataset title | Dataset URL | Database and Identifier |
|---|---|---|---|---|
| Xu H, Li Z, Kuo C, Götz K, Look T, Toledo MA, Seré K, Costa IG, Zenke M | 2023 | A lncRNA identifies Irf8 enhancer element in negative feedback control of dendritic cell differentiation | https://www.ncbi.nlm.nih.gov/geo/query/acc.cgi?acc=GSE198651 | NCBI Gene Expression Omnibus, GSE198651 |
| Xu H, Li Z, Kuo CC, Goetz K, Look T, de Toldeo MAS, Sere K, Costa IG, Zenke M | 2022 | Mus musculus lncIRF8 lncRNA, partial sequence | https://www.ncbi.nlm.nih.gov/nuccore/ON134061 | NCBI Nucleotide, ON134061 |
| Xu H, Li Z, Kuo CC, Goetz K, Look T, de Toldeo MAS, Sere K, Costa IG, Zenke M | 2022 | Mus musculus TCONS_00190258 lncRNA, partial sequence | https://www.ncbi.nlm.nih.gov/nuccore/ON134062 | NCBI Nucleotide, ON134062 |

The following previously published datasets were used:

| Author(s) | Year | Dataset title | Dataset URL | Database and Identifier |
|---|---|---|---|---|
| Li et al | 2019 | Identification of Transcription Factor Binding Sites using ATAC-seq | https://www.ncbi.nlm.nih.gov/geo/query/acc.cgi?acc=GSE118221 | NCBI Gene Expression Omnibus, GSE118221 |
| Garber et al | 2012 | A high throughput in vivo protein-DNA mapping approach reveals principles of dynamic gene regulation in mammals (ChIP-Seq) | https://www.ncbi.nlm.nih.gov/geo/query/acc.cgi?acc=GSE36099 | NCBI Gene Expression Omnibus, GSE36099 |
| Allhoff et al | 2016 | Differential Peak Calling of ChIP-Seq Signals with Replicates with THOR | https://www.ncbi.nlm.nih.gov/geo/query/acc.cgi?acc=GSE73143 | NCBI Gene Expression Omnibus, GSE73143 |
| Allhoff et al | 2014 | Detecting differential peaks in ChIP-seq signals with ODIN | https://www.ncbi.nlm.nih.gov/geo/query/acc.cgi?acc=GSE57563 | NCBI Gene Expression Omnibus, GSE57563 |
| Lin et al | 2015 | Epigenetic Program and Transcription Factor Circuitry of Dendritic Cell Development | https://www.ncbi.nlm.nih.gov/geo/query/acc.cgi?acc=GSE64767 | NCBI Gene Expression Omnibus, GSE64767 |
| Rodrigues et al | 2018 | A distinct lineage of origin reveals heterogeneity of plasmacytoid dendritic cells II (scRNAseq) | https://www.ncbi.nlm.nih.gov/geo/query/acc.cgi?acc=GSE114313 | NCBI Gene Expression Omnibus, GSE114313 |
| Pang et al | 2022 | Bulk RNAseq of Mouse Splenic Dendritic Cell Subsets | https://www.ncbi.nlm.nih.gov/geo/query/acc.cgi?acc=GSE188992 | NCBI Gene Expression Omnibus, GSE188992 |

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

# Appendix 1

## Appendix 1—key resources table

| Reagent type (species) or resource | Designation | Source or reference | Identifiers | Additional information |
|---|---|---|---|---|
| Genetic reagent (*M. musculus*) | C57BL/6 mice (CD45.1 and CD45.2) | Jackson Laboratory | | |
| Genetic reagent (*M. musculus*) | Mx-Cas9-GFP mice | Medical Faculty, RWTH Aachen University *Xu et al., 2022* | | |
| Cell line (*H. sapiens*) | HEK293T | ATCC | https://www.atcc.org | Lentivirus and retrovirus production |
| Cell line (*M. musculus*) | *lncIrf8* promoter KO HoxB8 MPP; Control HoxB8 MPP | This paper | | *lncIrf8* promoter KO |
| Cell line (*M. musculus*) | p*lncIrf8*-pA HoxB8 MPP; pGFP-pA HoxB8 MPP | This paper | | *lncIrf8* overexpression and rescue |
| Cell line (*M. musculus*) | *Irf8*-VP64 HoxB8 MPP; *lncIrf8*-VP64 HoxB8 MPP; Non-targeting-VP64-1 HoxB8 MPP; Non-targeting-VP64-2 HoxB8 MPP | This paper | | *Irf8* and *lncIrf8* promoters activation |
| Cell line (*M. musculus*) | *Irf8*-KRAB HoxB8 MPP; *lncIrf8*-KRAB HoxB8 MPP; Non-targeting-KRAB-1 HoxB8 MPP; Non-targeting-KRAB-2 HoxB8 MPP | This paper | | *Irf8* and *lncIrf8* promoters repression |
| Antibody | APC/Cyanine 7 anti-mouse/human B220 (rat monoclonal) | Biolegend | Cat# 103223; RRID: AB_313006 | FACS (1:400) |
| Antibody | Brilliant Violet 510 anti-mouse/human CD11b (rat monoclonal) | Biolegend | Cat# 101245; RRID: AB_2561390 | FACS (1:400) |
| Antibody | PE/Cyanine 7 anti-mouse CD11c (Armenian hamster monoclonal) | Biolegend | Cat# 117317; RRID: AB_493569 | FACS (1:400) |
| Antibody | APC anti-mouse CD115 (rat monoclonal) | eBioscience | Cat# 17-1152-80; RRID: AB_1210789 | FACS (1:400) |
| Antibody | PE/Cyanine 7 anti-mouse CD117 (rat monoclonal) | eBioscience | Cat# 25-1172-82; RRID: AB_469646 | FACS (1:400) |
| Antibody | PE anti-mouse CD135 (rat monoclonal) | eBioscience | Cat# 12-1351-82; RRID: AB_465859 | FACS (1:400) |
| Antibody | Biotin anti-mouse CD19 (rat monoclonal) | Biolegend | Cat# 115503; RRID: AB_313638 | FACS (1:800) |
| Antibody | Biotin anti-mouse CD3e (Armenian hamster monoclonal) | eBioscience | Cat# 13-0031-82; RRID: AB_466319 | FACS (1:800) |
| Antibody | APC/Cyanine 7 anti-mouse CD45.2 (mouse monoclonal) | Biolegend | Cat# 109823; RRID: AB_830788 | FACS (1:400) |
| Antibody | Biotin anti-mouse F4/80 (rat monoclonal) | Biolegend | Cat# 123105; RRID: AB_893499 | FACS (1:800) |
| Antibody | Alexa Fluor 700 anti-mouse Gr1 (rat monoclonal) | Biolegend | Cat# 108421; RRID: AB_493728 | FACS (1:400) |
| Antibody | PerCP/Cyanine 5.5 anti-mouse Gr1 (rat monoclonal) | eBioscience | Cat# 45-5931-80; RRID: AB_906247 | FACS (1:400) |
| Antibody | Alexa Fluor 700 anti-mouse Ly6C (rat monoclonal) | Biolegend | Cat# 128023; RRID: AB_10640119 | FACS (1:400) |

*Appendix 1 Continued on next page*

*Appendix 1 Continued*

| Reagent type (species) or resource | Designation | Source or reference | Identifiers | Additional information |
|---|---|---|---|---|
| Antibody | Biotin anti-mouse Ly6G (rat monoclonal) | Biolegend | Cat# 127603; RRID: AB_1186105 | FACS (1:800) |
| Antibody | Brilliant Violet 785 anti-mouse MHCII (rat monoclonal) | Biolegend | Cat# 107645; RRID: AB_2565977 | FACS (1:2000) |
| Antibody | Biotin anti-mouse NK1.1 (mouse monoclonal) | eBioscience | Cat# 14-5941-82; RRID: AB_467736 | FACS (1:800) |
| Antibody | Super Bright anti-mouse Siglec-H (rat monoclonal) | Invitrogen | Cat# 63-0333-82 RRID: AB_2784853 | FACS (1:400) |
| Antibody | PE/Dazzle 594 Streptavidin | Biolegend | Cat# 405247 | FACS (1:1000) |
| Antibody | Biotin anti-mouse Ter119 (rat monoclonal) | eBioscience | Cat# 14-5921-82; RRID: AB_467727 | FACS (1:800) |
| Antibody | Brilliant Violet 421 anti-mouse/rat XCR1 (mouse monoclonal) | Biolegend | Cat# 148216; RRID: AB_2565230 | FACS (1:400) |
| Antibody | 7-Aminoactinomycin D (7-AAD) | Thermo Fisher Scientific | Cat# A1310 | FACS (3 µl per test) |
| Chemical compound, drug | β-estradiol (E2) | Sigma-Aldrich | Cat# E2758 | |
| Chemical compound, drug | β-mercaptoethanol (β-ME) | Gibco | Cat# 31350010 | |
| Chemical compound, drug | BsmBI-v2 | New England Biolabs | Cat# R0739S | |
| Chemical compound, drug | Chondroitin sulfate sodium salt from shark cartilage (CSS) | Sigma-Aldrich | Cat# C4384 | |
| Chemical compound, drug | cOmplete Mini | Roche | Cat# 11836153001 | |
| Chemical compound, drug | dATP | New England Biolabs | Cat# N0440S | |
| Chemical compound, drug | Dimethysulfoxide (DMSO) | Sigma-Aldrich | Cat# D8418 | |
| Chemical compound, drug | Doxycycline hyclate | Sigma-Aldrich | Cat# D9891 | |
| Chemical compound, drug | DpnII | A kind gift from A. Marieke Oudelaar, Max Planck Institute for Multidisciplinary Sciences, Göttingen, Germany. DpnII enzyme with a similar activity is also available from New England Biolabs. | Cat# R0543M | |
| Chemical compound, drug | DMEM | Gibco | Cat# 41965039 | |
| Chemical compound, drug | DTT | Thermo Fisher Scientific | Cat# 20290 | |
| Chemical compound, drug | EDTA | Gibco | Cat# 15575–038 | |
| Chemical compound, drug | Fetal calf serum (FCS) | PAA | Cat# A01125-499 | |
| Chemical compound, drug | Fetal calf serum (FCS) | Gibco | Cat# 10270106 | |
| Chemical compound, drug | Formaldehyde (37%) | AppliChem | Cat# A0877 | |
| Chemical compound, drug | Recombinant human Flt3-Ligand (Flt3L) | Peprotech | Cat# 300–19 | |

*Appendix 1 Continued*

| Reagent type (species) or resource | Designation | Source or reference | Identifiers | Additional information |
|---|---|---|---|---|
| Chemical compound, drug | Recombinant murine stem cell factor (SCF) | Peprotech | Cat# 250–03 | |
| Chemical compound, drug | Human IGF-1 long range | Sigma-Aldrich | Cat# 85,580 C | |
| Chemical compound, drug | Recombinant IL-6/soluble IL-6 receptor fusion protein | A kind gift from S. Rose-John, Kiel, Germany *Fischer et al., 1997*. R&D Systems provides a similar product with the same activity. | Cat# 9038 SR | |
| Chemical compound, drug | HEPES | Sigma-Aldrich | Cat# H4034 | |
| Chemical compound, drug | L-glutamine | Gibco | Cat# 25030081 | |
| Chemical compound, drug | M-270 Streptavidin Dynabeads | Invitrogen | Cat# 65305 | |
| Chemical compound, drug | Mouse interferon α (mIFNα) | Miltenyi Biotec | Cat# 130-093-131 | |
| Chemical compound, drug | Murine RNase Inhibitor | New England Biolabs | Cat# M0314S | |
| Chemical compound, drug | Pancoll human, density 1.077 g/ml (Ficoll) | PAN-Biotech | Cat# P04-601000 | |
| Chemical compound, drug | Penicillin/streptomycin | Gibco | Cat# 15140122 | |
| Chemical compound, drug | Phenol-Chloroform-Isoamyl alcohol (PCI) | Sigma-Aldrich | Cat# 77617 | |
| Chemical compound, drug | Phosphate buffered saline (PBS) | Gibco | Cat# 10010023 | |
| Chemical compound, drug | Polybrene (PB, Hexadimethrine bromide) | Sigma-Aldrich | Cat# H9268 | |
| Chemical compound, drug | Q5 high fidelity DNA polymerase | New England Biolabs | Cat# M0491L | |
| Chemical compound, drug | RPMI 1640 | Gibco | Cat# 31870025 | |
| Chemical compound, drug | SalI | New England Biolabs | Cat# R0138S | |
| Chemical compound, drug | Supernatant from Flt3L-producing B16F1 cells (1%) | Homemade. Flt3L from Peprotech has the same activity (1:1000) | Cat# 300–19 | |
| Chemical compound, drug | Supernatant from CHO KLS C6 cells expressing soluble murine SCF (1%) | Homemade. Peprotech provides a similar product with the same activity (1:1000). | Cat# 250–03 | |
| Chemical compound, drug | SYBR-green fluorescence | Applied Biosystems | Cat# 4385610 | |
| Chemical compound, drug | T4 DNA HC ligase | Life Tech | Cat# EL0013 | |
| Chemical compound, drug | Taq DNA polymerase | Homemade | | |
| Chemical compound, drug | Taq buffer (10 x) | Thermo Fisher Scientific | Cat# B33 | |
| Chemical compound, drug | Template-switching RT enzyme mix | New England Biolabs | Cat# M0466 | |
| Chemical compound, drug | XhoI | New England Biolabs | Cat# R0146S | |

*Appendix 1 Continued on next page*

*Appendix 1 Continued*

| Reagent type (species) or resource | Designation | Source or reference | Identifiers | Additional information |
|---|---|---|---|---|
| Commercial assay or kit | Gibson Assembly kit | New England Biolabs | Cat# E5510S | |
| Commercial assay or kit | High-Capacity cDNA Reverse Transcription Kit | Applied Biosystems | Cat# 4368814 | |
| Commercial assay or kit | KAPA Hyper Capture Reagent Kit | Roche | Cat# 9075828001 | |
| Commercial assay or kit | NEBNext Ultra II DNA Library Prep Kit for Illumina | New England Biolabs | Cat# E7645S | |
| Commercial assay or kit | NEBNext Multiplex Oligos for Illumina (Index Primers Set 1) | New England Biolabs | Cat# E7335S | |
| Commercial assay or kit | NEBNext Multiplex Oligos for Illumina (Index Primers Set 2) | New England Biolabs | Cat# E7500S | |
| Commercial assay or kit | NucleoSpin RNA kit | Macherey-Nagel | Cat# 740955.250 | |
| Commercial assay or kit | PCR clean-up kit | Macherey-Nagel | Cat# 740609.50 | |
| Commercial assay or kit | TA cloning kit | Thermo Fisher Scientific | Cat# K202020 | |
| Sequence-based reagent | 5'RACE-TSO | New England Biolabs | 5' RACE PCR primers | GCTAATCATTGCAAGCAGTGGTATC AACGCAGAGTACATrGrGrG |
| Sequence-based reagent | 5'RACE-TSO-Specific | New England Biolabs | 5' RACE PCR primers | CATTGCAAGCAGTGGTATCAAC |
| Sequence-based reagent | 5'RACE-GSP-*lncIrf8*-R1 | New England Biolabs | 5' RACE PCR primers | TGTCAGTGATGGGGGCTGGAGAAAT |
| Sequence-based reagent | 5'RACE-GSP-*lncIrf8*-R2 | New England Biolabs | 5' RACE PCR primers | GCTCAGGATGCCAGGTCCCTTCTT |
| Sequence-based reagent | 3'RACE-QT | *Scotto-Lavino et al., 2006* | 3' RACE PCR primers | CCAGTGAGCAGAGTGACGAGGACTC GAGCTCAAGCTTTTTTTTTTTTTTTTT |
| Sequence-based reagent | 3'RACE-Q0 | *Scotto-Lavino et al., 2006* | 3' RACE PCR primers | CCAGTGAGCAGAGTGACG |
| Sequence-based reagent | 3'RACE-QI | *Scotto-Lavino et al., 2006* | 3' RACE PCR primers | GAGGACTCGAGCTCAAGC |
| Sequence-based reagent | 3'RACE-GSP-*lncIrf8*-F1 | This paper | 3' RACE PCR primers | ATTTCTCCAGCCCCCATCACTGACA |
| Sequence-based reagent | 3'RACE-GSP-*lncIrf8*-F2 | This paper | 3' RACE PCR primers | AAGAAGGGACCTGGCATCCTGAGC |
| Sequence-based reagent | *lncIrf8*-F | This paper | Genotyping primers | TCCTGAAGGGACAGGCAAG |
| Sequence-based reagent | *lncIrf8*-R | This paper | Genotyping primers | CTTGGACATTGAGGACGCC |
| Sequence-based reagent | cDC1 +32 kb-F1 | This paper | Genotyping primers | GTGACTGCAAGTAAGTTCTTCGG |
| Sequence-based reagent | cDC1 +32 kb-F2 | This paper | Genotyping primers | AAGTAGAGATTCCCTTTCTAAGCC |
| Sequence-based reagent | cDC1 +32 kb-R | This paper | Genotyping primers | ATCAGGCTGGGTGGTGGTT |
| Sequence-based reagent | Sc-*lncIrf8*-F | This paper | Cloning primers | ACACTCGAGACTGTCAGATGCAGGGG; the underline sequences represent cloning sites |
| Sequence-based reagent | Sc-*lncIrf8*-R | This paper | Cloning primers | AAAAAAGTCGACGCATCAGATTTAATATA GAACTAGGACA; the underline sequences represent cloning sites |
| Sequence-based reagent | CMV-*lncIrf8*-F | This paper | Genotyping primers | TGGGCGTGGATAGCGGTTT |

*Appendix 1 Continued on next page*

*Appendix 1 Continued*

| Reagent type (species) or resource | Designation | Source or reference | Identifiers | Additional information |
|---|---|---|---|---|
| Sequence-based reagent | CMV-*lncIrf8*-R | This paper | Genotyping primers | CACTGAGACTTAGCAAGGGGGA |
| Sequence-based reagent | CMV-GFP-F | This paper | Genotyping primers | TGGGCGTGGATAGCGGTTT |
| Sequence-based reagent | CMV-GFP-R | This paper | Genotyping primers | TGGGGGTGTTCTGCTGGTAG |
| Sequence-based reagent | m*lncIrf8*-tQ-F | This paper | RT-qPCR primers | ACTGTCAGATGCAGGGG |
| Sequence-based reagent | m*lncIrf8*-tQ-R | This paper | RT-qPCR primers | TCACAATCGTCTGTAACTCCG |
| Sequence-based reagent | m*Irf8*-tQ-F | This paper | RT-qPCR primers | GAGCGAAGTTCCTGAGATGG |
| Sequence-based reagent | m*Irf8*-tQ-R | This paper | RT-qPCR primers | TGGGCTCCTCTTGGTCATAC |
| Sequence-based reagent | m*GAPDH*-tQ-F | This paper | RT-qPCR primers | ACCTGCCAAGTATGATGACATCA |
| Sequence-based reagent | m*GAPDH*-tQ-R | This paper | RT-qPCR primers | GGTCCTCAGTGTAGCCCAAGAT |
| Sequence-based reagent | m3C-F | *Downes et al., 2021*; *Downes et al., 2022* | Capture-C qPCR primers | GGAGAAAGAAGGCTGGTGTTAT |
| Sequence-based reagent | m3C-R | *Downes et al., 2021*; *Downes et al., 2022* | Capture-C qPCR primers | TATCTGAGTTGGACAGCATTGG |
| Sequence-based reagent | m3C-control-F | *Downes et al., 2021*; *Downes et al., 2022* | Capture-C qPCR primers | TTATCTTGCATTTGCCAACTCG |
| Sequence-based reagent | m3C-control-R | *Downes et al., 2021*; *Downes et al., 2022* | Capture-C qPCR primers | TGGGTTTCCCTGATTCTGAAA |
| Sequence-based reagent | *Irf8*_P_L | This paper | Capture probes | GATCCGTGCATCACCAGCCTCC TTGACCTTAGGCAGACGCCCCA GCCCCCCGGCCATTTTTGGGGCAGCC |
| Sequence-based reagent | *Irf8*_P_R | This paper | Capture probes | CCAAATGAACAAACACCTCTCCC TTTAAAATCTGCCTGATGGCCAA CTTCATAATGAAGAGAAATAGATC |
| Sequence-based reagent | gRNA-1-F | This paper | gRNAs | <u>CACCG</u>TCCATTATACTAAGATACCC; the underline sequences represent cloning sites |
| Sequence-based reagent | gRNA-1-R | This paper | gRNAs | <u>AAAC</u>GGGTATCTTAGTATAATGGA<u>C</u>; the underline sequences represent cloning sites |
| Sequence-based reagent | gRNA-2-F | This paper | gRNAs | <u>CACC</u>GGTGCCGAGAAAGGACACGT; the underline sequences represent cloning sites |
| Sequence-based reagent | gRNA-2-R | This paper | gRNAs | <u>AAAC</u>ACGTGTCCTTTCTCGGCACC; the underline sequences represent cloning sites |
| Sequence-based reagent | gRNA-KO1-5'F | *Durai et al., 2019* | gRNAs | <u>CACC</u>GTTGTGATCTTTGAGGTAGA; the underline sequences represent cloning sites |
| Sequence-based reagent | gRNA-KO1-5'R | *Durai et al., 2019* | gRNAs | <u>AAAC</u>TCTACCTCAAAGATCACAAC; the underline sequences represent cloning sites |
| Sequence-based reagent | gRNA-KO1-3'F | *Durai et al., 2019* | gRNAs | <u>CACC</u>GAACTGGCCTGGGGCAGGTC; the underline sequences represent cloning sites |
| Sequence-based reagent | gRNA-KO1-3'R | *Durai et al., 2019* | gRNAs | <u>AAAC</u>GACCTGCCCCAGGCCAGTTC; the underline sequences represent cloning sites |
| Sequence-based reagent | gRNA-KO2-5'F | This paper | gRNAs | <u>CACC</u>GACATTCTGCACCCCAGTCA; the underline sequences represent cloning sites |
| Sequence-based reagent | gRNA-KO2-5'R | This paper | gRNAs | <u>AAAC</u>TGACTGGGGTGCAGAATGTC; the underline sequences represent cloning sites |
| Sequence-based reagent | gRNA-KO2-3'F | This paper | gRNAs | <u>CACCG</u>AGGATCGCACCTCACCTACT; the underline sequences represent cloning sites |
| Sequence-based reagent | gRNA-KO2-3'R | This paper | gRNAs | <u>AAAC</u>AGTAGGTGAGGTGCGATCCT<u>C</u>; the underline sequences represent cloning sites |

*Appendix 1 Continued on next page*

*Appendix 1 Continued*

| Reagent type (species) or resource | Designation | Source or reference | Identifiers | Additional information |
|---|---|---|---|---|
| Sequence-based reagent | gRNA-*Irf8*-VP64-F | This paper | gRNAs | <u>CACCG</u>ACGGTCGCGCGAGCTAATTG; the underline sequences represent cloning sites |
| Sequence-based reagent | gRNA-*Irf8*-VP64-R | This paper | gRNAs | <u>AAAC</u>CAATTAGCTCGCGCGACCGT<u>C;</u> the underline sequences represent cloning sites |
| Sequence-based reagent | gRNA-*Irf8*-KRAB-F | This paper | gRNAs | <u>CACC</u>GCGGCAGGTAGGACGCGATG; the underline sequences represent cloning sites |
| Sequence-based reagent | gRNA-*Irf8*-KRAB-R | This paper | gRNAs | <u>AAAC</u>CATCGCGTCCTACCTGCCGC; the underline sequences represent cloning sites |
| Sequence-based reagent | gRNA-*lncIrf8*-VP64-F | This paper | gRNAs | <u>CACC</u>GGTGCCGAGAAAGGACACGT; the underline sequences represent cloning sites |
| Sequence-based reagent | gRNA-*lncIrf8*-VP64-R | This paper | gRNAs | <u>AAAC</u>ACGTGTCCTTTCTCGGCACC; the underline sequences represent cloning sites |
| Sequence-based reagent | gRNA-*lncIrf8*-KRAB-F | This paper | gRNAs | <u>CACC</u>GAGTCACTCGTCCTTTGGGG; the underline sequences represent cloning sites |
| Sequence-based reagent | gRNA-*lncIrf8*-KRAB-R | This paper | gRNAs | <u>AAAC</u>CCCCAAAGGACGAGTGACTC; the underline sequences represent cloning sites |
| Sequence-based reagent | gRNA-non-targeting-1-F | *Manguso et al., 2017* | gRNAs | <u>CACCG</u>CGAGGTATTCGGCTCCGCG; the underline sequences represent cloning sites |
| Sequence-based reagent | gRNA-non-targeting-1-R | *Manguso et al., 2017* | gRNAs | <u>AAAC</u>CGCGGAGCCGAATACCTCG<u>C</u>; the underline sequences represent cloning sites |
| Sequence-based reagent | gRNA-non-targeting-2-F | *Manguso et al., 2017* | gRNAs | <u>CACCG</u>ATGTTGCAGTTCGGCTCGAT; the underline sequences represent cloning sites |
| Sequence-based reagent | gRNA-non-targeting-2-R | *Manguso et al., 2017* | gRNAs | <u>AAAC</u>ATCGAGCCGAACTGCAACAT<u>C;</u> the underline sequences represent cloning sites |
| Software, algorithm | Bamtofastq | 10 x Genomics | https://support.10xgenomics.com/docs/bamtofastq | |
| Software, algorithm | Bowtie2 | *Langmead and Salzberg, 2012* | http://bowtie-bio.sourceforge.net | |
| Software, algorithm | CapCruncher (v0.1.1a1) | *Downes et al., 2022* | https://github.com/sims-lab/CapCruncher | |
| Software, algorithm | CPAT | *Wang et al., 2013* | http://code.google.com/p/cpat/ | |
| Software, algorithm | Cufflinks (version 2.0) | *Trapnell et al., 2012* | http://cufflinks.cbcb.umd.edu/ | |
| Software, algorithm | FlowJo V10 | BD Biosciences | | |
| Software, algorithm | IGV browser | Broad Institute | | |
| Software, algorithm | MOODS | *Korhonen et al., 2009* | https://www.cs.helsinki.fi/group/pssmfind/ | |
| Software, algorithm | Oligo | *Oudelaar et al., 2020* | https://oligo.readthedocs.io/en/latest/index.html | |
| Software, algorithm | PhyloCSF | *Lin et al., 2011* | http://compbio.mit.edu/PhyloCSF | |
| Software, algorithm | Prism | GraphPad | | |
| Software, algorithm | Scanpy | *Wolf et al., 2018* | | |
| Software, algorithm | Star aligner (version 2.4) | *Dobin et al., 2013* | http://code.google.com/p/rna-star/ | |
| Software, algorithm | Snapgene | GSL Biotech | | |
| Software, algorithm | UMAP | *Becht et al., 2019* | | |

