## [Editor Report]

Authors provide valuable evidence identifying a lncRNA transcribed specifically in the pDC subtype from the +32Kb promoter region which is also the region for the enhancer for Irf8 specifically in the cDC1 subtype. With convincing methodology, they provide in-depth analysis about the possible role of lncIrf8, and its promoter region and cross-talk with Irf8 promoter to identify that it is not the lncIRF8 itself but its promoter region that is crucial for pDC and cDC1 differentiation conferring feedback inhibition of Irf8 transcription. The work will be of interest to immunologists working on immune cell development.

---

## [Decision Letter]

**Decision letter after peer review:**

Thank you for submitting your article "A lncRNA identifies *IRF8* enhancer element in negative feedback control of dendritic cell differentiation" for consideration by *eLife*. Your article has been reviewed by 2 peer reviewers, and the evaluation has been overseen by a Reviewing Editor and Satyajit Rath as the Senior Editor. The following individual involved in the review of your submission has agreed to reveal their identity: Stephen L Nutt (Reviewer #2).

Essential revisions:

1) Demonstrate the presence of lncRNA selectively in ex-vivo splenic pDC sub-type and confirm its expression in human pDCs.

2) To clarify the role of irf8 in pdc development in regards to published data comparing the lncIrf8 and +32kb activity in their system: The authors conclude that lncIRF8 promoter KO hematopoietic progenitors are not capable of differentiating to pDCs (Figure 2C, Figure 2 Suppl 3D). However, it is now understood that Irf8 is not essential for pDC development (see Sichien Immunity 2016 where they study WT:IRF8-/- BM chimeras) and that Irf8 deregulates some key markers, including CD11b. The authors need to account for this discrepancy. Are pDCs with aberrant marker expression produced in the lncIRF8 promoter KO, and if not, why does this phenotype differ from the Irf8 KO? In addition, it is essential that authors delete the +32 Kb enhancer region (as defined in Durai 2019) in their culture system and analyze the same with respect to +32Kb lncIRF8 promoter deletion in terms of expression of IRF8, lncIRF8 and its effect on the development of pDC vs cDC1.

3)( Provide evidence that IRF8 binds in the lncIRF8 promoter. Authors showed show DNA binding motifs for Irf8 in the lncIrf8, but no evidence of actual binding. Perhaps the +32kb enhancer is also physically involved, but either way more evidence is needed to fully support the authors' models. Instead, IRF8 binds in the more 5' part of the +32kb enhancer identified by Durai 2019. How then is a putative IRF8 repressor complex linked to the lncIRF8 promoter as the authors propose? Moreover, the model does not appear to address why the IRF8 repressor complex binding to the lncIRF8 would be active in cDC1 and not pDC.

4) Authors should acknowledge/discuss clearly that one limitation of their study is that lncIRF8 might have other roles in DC/pDC biology which is yet unidentified in the current study.

*Reviewer #1 (Recommendations for the authors):*

1. While refining the cross-talk between promoter and enhancers authors have completely overlooked the functional significance of lncIRF8 in pDCs. Authors have demonstrated the expression of lncIRF8 is specific to pDCs, it is not evident whether all pDCs express it or a fraction of the same.

a. Further authors should confirm the expression of lncRNA in ex vivo pDCs from various murine tissues. Is the expression of lncRNA also conserved in human pDCs?

b. How is the expression of lncIRF8 modulated upon CpG stimulation/viral infection of pDCs?

c. Is there any significance of lncIRF8 in pDC functions (TLR stimulations, cytokine production, etc.) these aspects could be studied without much effort as authors already have systems that can knockout lncIRF8 promoter or overexpress lncIRF8.

2. Though authors have identified that lncIRF8 is specifically expressed in pDC subtype but while understanding its significance and regulation, it was preferred to study bulk DC cultures overlooking the finding regarding specificity. Experiments performed by employing dCas9 VP64/KRAB-GFP constructs, do provide some direction in terms of mechanistic understanding of the cross-talk between the promoter of IRF8 and lncIRF8. This data needs to be further strengthened by sorting pDC, cDC1 and cDC2 populations and studying the effect of the genetic manipulations on IRF8 and lncRNA expression in each cell type.

3. One of the most important findings of the submitted study is to demonstrate that knocking out the lncIRF8 promoter also diminishes the expression of IRF8, pDC as well as cDC1 development (Figure 2). Once the IRF8 expression is suppressed the analysis of pDC and cDC1 differentiation is redundant. It is previously reported that the deletion of +32 Kb enhancer (Durai et al. 2019) in a mouse model did not affect pDC but specifically depleted cDC1 development. Authors should examine the deletion of only the +32 kb enhancer region in their culture system to sort out the hierarchy between the +32 Kb enhancer and +32 Kb lncRNA promoter in terms of DC subtype development (cDC1 Vs pDC). Does the deletion of the lncIRF8 promoter lead to epigenetic changes at the overall +32 region? It is expected that due to interactions and chromatin loop formation between IRF8 promoter with various enhancers elements; deletion of lncIRF8 promoter might play an important role in defining these interactions in a cell type-specific manner. These interactions become more valuable by the fact that activation of the lncIRF8 promoter leads to enhanced cDC1 differentiation. Does this simply mean more open chromatin conformation around the cDC1-specific +32 Kb region rather than correlating with the lncIRF8?

Can authors analyze the epigenetic changes by performing similar experiments in some of the available pDC and cDC1 cell lines? As these cell lines are likely to have active IRF8 transcription and it would help to measure the effect of lncIRF8 promoter deletion/mutation/activation/repression on IRF8 expression and specific DC subtype maintenance.

4. Based on the various earlier published reports and results from the current study authors have shown a schematic representation of the pDC and cDC1-specific interactions. Can these interactions be validated in primary/ex-vivo DCs and be correlated with lncIRF8 as well as IRF8 expression? Will it be possible for authors to validate these interactions in light of the lncIRF8 promoter deletion/mutation/activation/repression? Authors may be careful to perform these experiments in purified pDC and cDC1 fractions to get meaningful data.

In case it is not possible to perform experiments with primary DCs/genetically modified HOXB8 cultures; if required, some of these experiments may be performed in the cell lines wherein overall interaction between various enhancer elements and promoters can be validated.

5. Authors have also identified a TCONS_00190258 which is specifically expressed in the cDC1 subtype. Why not look at the expression levels of this transcript in the various experiments in the current study as enhancer element crosstalk will help in cell type-specific transcription of IRF8, lncIRF8, and TCONS_00190258?

*Reviewer #2 (Recommendations for the authors):*

I find the quality and interpretation of the genomics data to be excellent and I have no major concerns with that data. I do have some specific comments around the interpretation of the cellular data and the model presented.

Specific comments

1. The authors conclude that lncIRF8 promoter KO hematopoietic progenitors are not capable of differentiating to pDCs (Figure 2C, Figure 2 Suppl 3D). However, it is now understood that Irf8 is not essential for pDC development (see Sichien Immunity 2016 where they study WT:IRF8-/- BM chimeras) and that Irf8 deregulates some key markers, including CD11b. The authors need to account for this discrepancy. Are pDCs with aberrant marker expression produced in the lncIRF8 promoter KO, and if not, why does this phenotype differ from the Irf8 KO?

2. There does not appear to be compelling evidence that IRF8 binds in the lncIRF8 promoter in cDC1. Instead, IRF8 binds in the more 5' part of the +32kb enhancer identified by Durai 2019. How then is a putative IRF8 repressor complex linked to the lncIRF8 promoter as the authors propose? Moreover, the model does not appear to address why the IRF8 repressor complex binding to the IRF8 repressor complex would be active in cDC1 and not pDC.

3. The authors use 2 versions of the HoxB8 MPP differentiation system (lines 207-234). Firstly, it is unclear, why two similar approaches are used, but more importantly, the second system outlined in figure 2 Suppl 3 shows that the lncIRF8 promoter is also required for cDC2 differentiation. This is a concern, as IRF8 is not expressed in cDC2, and Irf8-/- mice produce cDC2.

---

## [Author Response]

Essential revisions:1) Demonstrate the presence of lncRNA selectively in ex-vivo splenic pDC sub-type and confirm its expression in human pDCs.

Point 1: Presence of lncRNA selectively in ex-vivo splenic pDC sub-type.

This is interesting and we addressed this point by re-analyzing public scRNA-seq and bulk RNA-seq datasets for spleen. We also included the analysis of lncRNA expression in bone marrow. The scRNA-seq data of murine *ex-vivo* spleen pDC and bone marrow pDC were obtained from GSE114313 (Rodrigues et al., *Nat. Immunol.,* 2018). The bulk RNA-seq dataset of splenic DC subsets pDC, cDC1 and cDC2 are from GSE188992 (Pang et al., *Front. Immunol.*, 2022).

These new results are now shown in new Figure 1—figure supplement 3 (page 32) and described in the Results section, page 6, lines 151-154 as follows:

*“lncIrf8* and *TCONS_00190258* show the same expression pattern in pDC and cDC1, respectively, in BM and spleen (Figure 1—figure supplement 3), as revealed by reanalyzing scRNA-seq and bulk RNA-seq data (*Rodrigues et al., 2018; Pang et al., 2022*).”

The Materials and methods section was updated accordingly (page 17, lines 536-548).

More specifically, the scRNA-seq data from GSE114313 (Rodrigues et al., *Nat. Immunol.,* 2018) were downloaded, BAM files were converted back into FASTQ files and a custom reference genome of mm9 was created now including the novel *lncIrf8* (GenBank: ON134061) and *Tcons_00190258* lncRNA (GenBank: ON134062). Data were subjected to bioinformatic processing as described in the Materials and methods section on page 17, lines 536-548 and visualized in UMAPs.

In Dress et al., *Nat. Immunol*., 2019 authors are critical on the pDC-like cells in GSE114313. Therefore to better define the pDC population in GSE114313, we included the pDC marker genes *Tcf4*, *Bst2* and *Siglec-H* in our analysis and used the cDC marker genes *Zbtb46*, *Cx3cr1* and *Xcr1* as negative controls. We observed that *lncIrf8* is expressed in bone marrow (BM) and spleen pDC (new Figure 1—figure supplement 3, page 32)*.* Expression of the cDC1 specific *Tcons_00190258* lncRNA in BM and spleen pDC is very low to absent.

To further extend this analysis, we analyzed the bulk RNA-seq dataset of splenic DC subsets pDC, cDC1 and cDC2 from GSE188992 (Pang et al., *Front. Immunol.*, 2022). *lncIrf8* and *Tcons_00190258* lncRNA were visualized by IGV browser. As expected, *lncIrf8* and *Tcons_00190258* lncRNA were expressed in pDC and cDC1 of spleen, respectively, and the expression pattern was the same as in our in vitro bone marrow DC (new Figure 1—figure supplement 3).

Point 2: Confirm its expression in human pDCs.

To address this point, we checked the transcripts around the *Irf8* gene in different species using data from NCBI, including mouse, human and zebrafish. We found a lncRNA downstream of human *Irf8* gene, referred to as *LINC02132*, but no lncRNA was observed downstream of zebrafish *Irf8* gene (see Author response image 1).

**Author response image 1. sa2fig1:** Mouse *lncIrf8* is not conserved across species. (A) Visualization of lncRNA downstream of *Irf8* gene in mouse, human and zebrafish using data from NCBI. Red box indicates lncRNA. (B) Sequence alignment of mouse *lncIrf8* and human *LINC02132* using MEGA 6.0. (C) Visualization of human *LINC02132* by UCSC browser. Data of gene transcription, H3K27ac signal and DNase signal are from ENCODE. (D) Gene expression of human *LINC02132* in different tissues. Data are from GTEx RNA-seq in 54 tissues of 17382 samples, 948 donors.

We found that the sequences of mouse *lncIrf8* and human *LINC02132* are not conserved (Author response image 1). The human *LINC02132* locus also shows potential enhancer activities, as revealed by H3K27ac and open chromatin (DNase) signals (Author response image 1). The human *LINC02132* is mainly expressed in lymphocytes, spleen and whole blood (Author response image 1). Data of H3K27ac (7 cell lines) and DNase signal are from ENCODE; gene expression of human *LINC02132* are from GTEx RNA-seq (54 tissues, 17382 samples, 948 donors).

Taken together, the pDC specific *lncIrf8* is not conserved across species, which is in line with the general characteristics of lncRNA. We have now included this information in the Results section, page 7, lines 200-202.

2) To clarify the role of irf8 in pdc development in regards to published data comparing the lncIrf8 and +32kb activity in their system: The authors conclude that lncIRF8 promoter KO hematopoietic progenitors are not capable of differentiating to pDCs (Figure 2C, Figure 2 Suppl 3D). However, it is now understood that Irf8 is not essential for pDC development (see Sichien Immunity 2016 where they study WT:IRF8-/- BM chimeras) and that Irf8 deregulates some key markers, including CD11b. The authors need to account for this discrepancy. Are pDCs with aberrant marker expression produced in the lncIRF8 promoter KO, and if not, why does this phenotype differ from the Irf8 KO? In addition, it is essential that authors delete the +32 Kb enhancer region (as defined in Durai 2019) in their culture system and analyze the same with respect to +32Kb lncIRF8 promoter deletion in terms of expression of IRF8, lncIRF8 and its effect on the development of pDC vs cDC1.

Point 1: Are pDCs with aberrant marker expression produced in the lncIRF8 promoter KO?

We appreciate the point raised and know the work by Sichien et al., 2016 on WT:*Irf8-/-* BM chimeras and the conclusions reached from their work that IRF8 is dispensable for pDC development. Previous studies, including the study of Sichien et al., 2016, showed that pDC are reduced in *Irf8* knockout mice (Murakami et al., *Nat. Immunol.,* 2021; Schiavoni et al., *J. Exp. Med.,* 2002; Tsujimura et al., *J. Immunol.,* 2003). Thus, there is the possibility that the conclusions reached by Sichien et al., 2016 on the impact of IRF8 on pDC function, is on the pDC population, which is left in the *Irf8-/-* mice. In addition the results in Sichien et al., 2016 are based on WT:*Irf8-/-* BM chimeras upon transplantation in lethally irradiated mice and thus the divergent results might be due to the experimental set-up WT:*Irf8-/-* BM chimeras versus *Irf8* knockout mice.

Sichien et al., 2016, report on altered marker expression (CD11b, CD11c and MHC II), which for CD11b and CD11c was not very prominent. In our study on the *lncIrf8* promoter KO we used immunophenotyping of CD11c+ CD11b- B220+ cells for in vitro pDC (Figure 2C and Figure 2—figure supplement 3) and CD45.2+ Gr1- Siglec H^+^ cells for pDC following transplantation (Figure 3A and D, G and J; Figure 3—figure supplement 1B). Thus, we are confident to correctly detect pDC.

A potentially altered DC marker expression on *lncIrf8* promoter KO pDC cannot be measured because cells are absent in *lncIrf8* promoter KO. Nevertheless, we visualized CD11c and CD11b expression on cells which are present by gating on single cells on day 7 and 9 of Flt3L directed DC differentiation in *lncIrf8* promoter KO and control cultures. In *lncIrf8* promoter KO cells CD11c expression was slightly reduced and CD11b expression was slightly increased (see Author response image 2). This indicates that *lncIrf8* promoter KO does not have a major effect on the expression of CD11c and CD11b. Given this negative result we prefer not to include these data in the manuscript.

**Author response image 2. sa2fig2:** CD11c and CD11b median fluorescence intensity (MFI) of single cells in *lncIrf8* promoter KO and Control cells.

Point 2: Deletion of the +32 kb enhancer region as defined in Durai et al. 2019.

We followed the request of the reviewers and deleted the +32 kb enhancer region by using the same gRNAs as described in Durai et al., *Nat. Immunol*., 2019, referred to as cDC1 specific +32 kb enhancer KO (see new Figure 2—figure supplement 4C, page 35). In parallel, we also designed a new pair of gRNAs to delete this +32 kb enhancer region. We followed the same procedure as in the *lncIrf8* promoter KO experiments (Figure 2—figure supplement 1A) for both the Durai et al. gRNAs and our new self-designed gRNAs to obtain homozygous clones of cDC1 specific +32 kb enhancer KO. Five out of 165 single-cell clones were subjected to DC differentiation and further analyzed at day 7 and 9. Frequencies of pDC were unaffected by +32 kb enhancer KO at the end of culture (day 9). There was some effect at day 7, however this was not statistically significant. The cDC1 specific +32 kb enhancer KO compromised cDC1 development at day 7 and day 9, which is in line with the phenotypes described by Durai et al., *Nat. Immunol*., 2019 and Murakami et al., *Nat. Immunol*., 2021 in mice.

These new results are now shown in new Figure 2—figure supplement 4 in the revised manuscript (Results section, page 8, lines 235-249 and page 35). The Materials and methods section was also updated accordingly (page 18, lines 576-605). Source data of new Figure 2—figure supplement 4B are also provided and described on page 36 (Figure 2—figure supplement 4-source data 1 and source data 2)

In summary, our data show that deletion of cDC1 specific +32 kb enhancer affects cDC1 development while the *lncIrf8* promoter KO affects both pDC and cDC1 development (Figure 2). This indicates that the *lncIrf8* promoter element has further functions compared to the cDC1 specific +32 kb enhancer, e.g. to control pDC development. We have now indicated this point in the Discussion (Discussion section, page 13, lines 422-427).

3) Provide evidence that IRF8 binds in the lncIRF8 promoter. Authors showed show DNA binding motifs for Irf8 in the lncIrf8, but no evidence of actual binding. Perhaps the +32kb enhancer is also physically involved, but either way more evidence is needed to fully support the authors' models. Instead, IRF8 binds in the more 5' part of the +32kb enhancer identified by Durai 2019. How then is a putative IRF8 repressor complex linked to the lncIRF8 promoter as the authors propose? Moreover, the model does not appear to address why the IRF8 repressor complex binding to the lncIRF8 would be active in cDC1 and not pDC.

We appreciate the reviewers’ point and can very well agree with the reviewers that the cDC1 specific +32 kb enhancer element by Durai et al. 2019 might also be involved in the negative feedback regulation of *Irf8* during DC differentiation. The full +32 kb enhancer is delineated by the H3K27ac and H3K4me3 marks (Figure 2A and new Figure 2—figure supplement 4A) and comprises both the +32 kb enhancer element by Durai et al. 2019 and the *lncIrf8* promoter described here. Both elements appear to be important for full *Irf8* +32 kb enhancer function in DC differentiation: (i) Deletion of the *lncIrf8* promoter compromised both pDC and cDC1 differentiation (Figure 2); (ii) Deletion of the +32 kb enhancer element by Durai et al. 2019 compromised cDC1 and to some extent also pDC differentiation (new Figure 2—figure supplement 4; although not statistical significant); (iii) positioning the dCas9-VP64 at the *lncIrf8* promoter for CRISPR activation prominently enhanced cDC1 differentiation (Figure 5A, B and E) and (iv) positioning the dCas9-KRAB at the *lncIrf8* promoter for CRISPR interference compromised both pDC and cDC1 differentiation (Figure 6A-E).

In our model we proposed that binding of the putative IRF8 repressor to the +32 kb enhancer is mainly in the 5’ part of the +32 kb enhancer, as demonstrated by the ChIP-seq of IRF8. There is a strong IRF8 binding signal detected in cDC and weak to no IRF8 binding signal in pDC (Figure 1A, Figure 1—figure supplement 1, Figure 2—figure supplement 2 and Figure 7B, ChIP-seq IRF8 signal).

Thus, we propose the putative IRF8 repressor complex is only active in cDC and not pDC (Figure 7B). We agree with the reviewers that at this stage of analysis we do not know why IRF8 binds to the 5’ part of the +32 kb enhancer in cDC but not in pDC. Addressing this point should be subject for further studies.

4) Authors should acknowledge/discuss clearly that one limitation of their study is that lncIRF8 might have other roles in DC/pDC biology which is yet unidentified in the current study.

We appreciate the suggestion made by the reviewer. We have now indicated that *lncIrf8* might have other roles in DC biology which is yet unidentified in the current study (Discussion section, page 13, lines 435 and 436).

Reviewer #1 (Recommendations for the authors):1. While refining the cross-talk between promoter and enhancers authors have completely overlooked the functional significance of lncIRF8 in pDCs. Authors have demonstrated the expression of lncIRF8 is specific to pDCs, it is not evident whether all pDCs express it or a fraction of the same.a. Further authors should confirm the expression of lncRNA in ex vivo pDCs from various murine tissues. Is the expression of lncRNA also conserved in human pDCs?

The question by the reviewer on the expression of the novel *lncIrf8* and *Tcons_00190258 lncRNA* in ex vivo pDC is well taken. We addressed this point be re-analyzing scRNA-seq data of BM and spleen pDC of GSE114313 (Rodrigues et al., *Nat. Immunol.,* 2018) and bulk RNA-seq data of the splenic DC subsets pDC, cDC1 and cDC2 of GSE188992 (Pang et al., *Front. Immunol.*, 2022).

The scRNA-seq data showed that *lncIrf8* is expressed in BM and spleen pDC. Expression of the cDC1 specific *Tcons_00190258* lncRNA in BM and spleen pDC is very low to absent. The bulk RNA-seq data fully support these results: *lncIrf8* and *Tcons_00190258* lncRNA are expressed in pDC and cDC1 of spleen, respectively, and the expression pattern is the same as in in vitro bone marrow DC.

These new data are in new Figure 1—figure supplement 3 *(*please see above our response to Essential Revision 1).

We also addressed the question on the conservation of the novel *lncIrf8* across species. A lncRNA downstream of the human *Irf8* gene was identified, referred to as *LINC02132*, but no lncRNA was observed downstream of zebrafish *Irf8* gene (see Author response image 1). By sequence alignment we demonstrate that the sequences of mouse *lncIrf8* and human *LINC02132* are not conserved (Author response image 1; see also above our response above to Essential Revision 1 for further details).

We conclude that the pDC specific *lncIrf8* is not conserved across species. This is a common characteristic, which is very frequently found for lncRNA. We now include this information in the Results section, page 7, lines 200-202.

b. How is the expression of lncIRF8 modulated upon CpG stimulation/viral infection of pDCs?

This is an interesting question and we addressed this point by re-analyzing public RNA-seq data of FACS sorted pDC (CD3- CD19- CD11c+ CD11blow B220+ Siglec-H^+^ CD317+) from BM cells of in vitro Flt3L cultures (GSE170750, Mann-Nuettel et al., *BMC Genom. Data*, 2021). In this study pDC were stimulated by CpG 2216 for 0, 2, 6 and 12 hours and subjected to RNA-seq analysis. Our analysis showed that *Irf8* expression was upregulated with time upon CpG stimulation, while *lncIrf8* expression was downregulated (Author response image 3). Obviously, pDC activation by CpG stimulation affects the expression of hundreds of transcription factors within a few hours (Author response image 3 taken from Mann-Nuettel et al., *BMC Genom. Data*, 2021), including IRF8. CpG stimulation of pDC occurs within hours and is different from pDC differentiation, which takes a few days.

**Author response image 3. sa2fig3:** Gene expression after pDC activation by CpG 2216 stimulation. (A) Visualization of *lncIrf8* and *Irf8* expression in BM cell-derived pDC upon CpG 2216 stimulation after 0, 2, 6 and 12 hours recalculated from RNA-seq data of FACS sorted pDC (CD3- CD19- CD11c+ CD11blow B220+ Siglec-H^+^ CD317+; GSE170750 from Mann-Nuettel et al., *BMC Genom. Data*, 2021). (B and C) Differentially expressed transcription factors in CpG 2216 stimulated pDC. Panel (A) is our calculation, panels (B) and (C) are from Mann-Nüttel et al., *BMC Genom. Data*, 2021 Figure 2B and Figure 2F.

Therefore, it is not easy to explain the gene expression patterns of *lncIrf8* and *Irf8* in pDC upon CpG stimulation and a potential cross talk between *lncIrf8* and *Irf8* using our model that builds on DC differentiation. Given these rather different scenarios and the complexity of both systems, we prefer to not include these data in the paper.

c. Is there any significance of lncIRF8 in pDC functions (TLR stimulations, cytokine production, etc.) these aspects could be studied without much effort as authors already have systems that can knockout lncIRF8 promoter or overexpress lncIRF8.

We understand the reviewer’s point on the interest in a potential significance of lncIRF8 on pDC functions using knockout or overexpression studies. However this point is difficult to address: (i) The *lncIrf8* promoter KO severely compromises pDC development and pDC are essentially absent in *lncIrf8* promoter KO cultures. (ii) lncIRF8 overexpression is difficult, as overexpression of lncRNA in general, which is challenging. This is because lncRNA overexpression in lentivirus vectors is very different from protein overexpression. To limit *lncIrf8* overexpression to the *lncIrf8* sequence a polyA signal for transcription termination has to be inserted at the 3 ’end of *lncIrf8* (Figure 4B). This polyA signal decrease the titer of lentivirus, which limits the overexpression capacity. We would expect to need a much higher *lncIrf8* overexpression than the one achieved here (Figure 4 and Figure 4—figure supplement 1) to reveal a potential significance of *lncIrf8* in pDC functions, such as TLR stimulations and/or cytokine production.

We are in accord with the reviewer that *lncIrf8* might have other roles in DC biology than the one identified in our study. We now included a sentence on the potential other roles of *lncIrf8* in DC biology, which are not yet identified in the current study (Discussion section, page 13, lines 435 and 436).

2. Though authors have identified that lncIRF8 is specifically expressed in pDC subtype but while understanding its significance and regulation, it was preferred to study bulk DC cultures overlooking the finding regarding specificity. Experiments performed by employing dCas9 VP64/KRAB-GFP constructs, do provide some direction in terms of mechanistic understanding of the cross-talk between the promoter of IRF8 and lncIRF8. This data needs to be further strengthened by sorting pDC, cDC1 and cDC2 populations and studying the effect of the genetic manipulations on IRF8 and lncRNA expression in each cell type.

We appreciate the reviewer’s suggestion on studying the effects of dCas9-VP64/KRAB-GFP constructs targeting the *lncIrf8* and *Irf8* promoters in sorted pDC, cDC1 and cDC2 populations. However, this is difficult and in some cases not possible, since the dCas9-VP64/KRAB-GFP constructs are introduced in HoxB8 multipotent progenitors (HoxB8 MPP) (Figure 5—figure supplement 1A) and following Flt3L directed DC differentiation some DC subsets are very low in abundance to even absent (Figures 5 and 6). For example, in the dCas9-KRAB experiment in Figure 6A-E, frequencies of cDC1 and pDC are very low to absent. In the dCas9-VP64 experiment in Figure 5 cDC1 and pDC are there, but the impact of targeted CRISPR activation on the *lncIrf8* promoter is mainly seen in HoxB8 MPP at day 0 of DC differentiation (Figure 5C) and the activation effect is largely lost in the differentiated cells at later time points. Thus, studying the impact of targeted CRISPR activation and interference on IRF8 and lncRNA expression in DC subsets should include also MPP/CDP and pre-DC, and should be ideally performed by scRNA-seq and scATAC-seq. This analysis is beyond the scope of the present study and will be subject of our future work.

3. One of the most important findings of the submitted study is to demonstrate that knocking out the lncIRF8 promoter also diminishes the expression of IRF8, pDC as well as cDC1 development (Figure 2). Once the IRF8 expression is suppressed the analysis of pDC and cDC1 differentiation is redundant. It is previously reported that the deletion of +32 Kb enhancer (Durai et al. 2019) in a mouse model did not affect pDC but specifically depleted cDC1 development. Authors should examine the deletion of only the +32 kb enhancer region in their culture system to sort out the hierarchy between the +32 Kb enhancer and +32 Kb lncRNA promoter in terms of DC subtype development (cDC1 Vs pDC). Does the deletion of the lncIRF8 promoter lead to epigenetic changes at the overall +32 region? It is expected that due to interactions and chromatin loop formation between IRF8 promoter with various enhancers elements; deletion of lncIRF8 promoter might play an important role in defining these interactions in a cell type-specific manner. These interactions become more valuable by the fact that activation of the lncIRF8 promoter leads to enhanced cDC1 differentiation. Does this simply mean more open chromatin conformation around the cDC1-specific +32 Kb region rather than correlating with the lncIRF8?

We agree with the reviewer that deletion of the *lncIrf8* promoter might lead to epigenetic changes in the entire +32 region and might have multiple effects, such as affecting transcription factor binding, histone modifications, open chromatin regions and chromatin interactions. Similarly, targeting dCas9-VP64 to the *lncIrf8* promoter for CRISPR activation caused more cDC1 differentiation, which might be also due to multiple effects, such as changes in transcription factor binding, chromatin interactions and/or chromatin conformation. Here CRISPR activation of the *lncIrf8* promoter might cause +32 kb enhancer activation and as a consequence IRF8 activation and more cDC1 (Figure 5A-E).

We now followed the request of the reviewer and deleted the +32 kb enhancer region with the four 4 AICE elements as described by Durai et al., *Nat. Immunol*., 2019 by using the same gRNAs as described in Durai et al., in the following referred to as cDC1 specific +32 kb enhancer KO. In parallel, we also designed a novel pair of gRNAs to delete this +32 kb enhancer region. We followed the same procedure as the *lncIrf8* promoter KO experiments (Figure 2—figure supplement 1A) to obtain homozygous clones of cDC1 specific +32 kb enhancer KO. Five out of 165 single-cell clones were subjected to DC differentiation and further analyzed. cDC1 specific +32 kb enhancer KO did not affect pDC differentiation but compromised cDC1 development, which is in line with the phenotypes described by Durai et al., *Nat. Immunol*., 2019 and Murakami et al., *Nat. Immunol*., 2021 in mice.

These new results are now shown in new Figure 2—figure supplement 4 in the revised manuscript (Results section, page 8, lines 235-249 and page 35). The Materials and methods section was also updated accordingly (page 18, lines 576-605). Please see above our response to Essential Revision 2.

Can authors analyze the epigenetic changes by performing similar experiments in some of the available pDC and cDC1 cell lines? As these cell lines are likely to have active IRF8 transcription and it would help to measure the effect of lncIRF8 promoter deletion/mutation/activation/repression on IRF8 expression and specific DC subtype maintenance.

We thank the reviewer for the suggestion to study the epigenetic changes reported here in some of the available pDC and cDC1 cell lines. However, we hesitate to do so, since established cell lines have inevitably suffered from mutation and multiple alterations, which are often unknown and/or hard to control. We rather have built our study on conditional immortalization of mouse bone marrow cells by HoxB8 (referred to as HoxB8 MPP; Figure 1—figure supplement 2A).

These conditionally immortalized HoxB8 MPP exhibit an essentially unlimited lifespan when grown with estrogen (E2), when HoxB8 is active (Xu et al., *Eur. J. Immunol.*, 2022). Additionally, they stop proliferating when E2 is removed and faithfully recapitulate hematopoietic stem cell commitment, DC differentiation and allow genetic manipulations by CRISPR/Cas tools. Thus, we consider HoxB8 MPP a particularly versatile and useful model to study gene regulations during DC differentiation. Repeating the study in pDC and cDC1 cell lines will, in our opinion, only provide limited new information but entails the risk of picking up aberrant phenomena, which are inherent to studies in cell lines.

4. Based on the various earlier published reports and results from the current study authors have shown a schematic representation of the pDC and cDC1-specific interactions. Can these interactions be validated in primary/ex-vivo DCs and be correlated with lncIRF8 as well as IRF8 expression? Will it be possible for authors to validate these interactions in light of the lncIRF8 promoter deletion/mutation/activation/repression? Authors may be careful to perform these experiments in purified pDC and cDC1 fractions to get meaningful data.In case it is not possible to perform experiments with primary DCs/genetically modified HOXB8 cultures; if required, some of these experiments may be performed in the cell lines wherein overall interaction between various enhancer elements and promoters can be validated.

We agree with the reviewer that validating the pDC and cDC1-specific interactions mediated by *lncIrf8* and *Irf8* in primary/ex-vivo DC should be interesting. In this context we now provide data that *ex-vivo* pDC express *lncIrf8* similar to in vitro BM pDC (new Figure 1—figure supplement 3), which indicates that the gene regulation patterns we found in our in vitro systems also exist in vivo.

Primary/*ex-vivo* DC have a limited lifespan in vitro, thus it will be challenging to impossible to generate genetically modified cells for studying gene regulations in primary cells. In particular, deletions of cis-regulatory elements (such as enhancer or promoter sequences) that require single-cell clones, are not possible with primary mouse cells due to lifespan restrictions. These limitations made us to develop the conditional immortalized mouse bone marrow HoxB8 MPP system (Figure 1—figure supplement 2A; Xu et al., *Eur. J. Immunol.*, 2022). Experimental details of the HoxB8 MPP-DC differentiation system are now included in a Guideline paper by Lutz et al., *Eur. J. Immunol.*, 2022. We added this reference to the reference list and cite this paper along with the Xu et al., 2022 paper as indicated.

5. Authors have also identified a TCONS_00190258 which is specifically expressed in the cDC1 subtype. Why not look at the expression levels of this transcript in the various experiments in the current study as enhancer element crosstalk will help in cell type-specific transcription of IRF8, lncIRF8, and TCONS_00190258?

We really appreciate the suggestion made by the reviewer. We have now provided evidence that *Tcons_00190258* is also express in *ex-vivo* tissue (new Figure 1—figure supplement 3C). Studying the impact of *Tcons_00190258* on DC differentiation and its potential crosstalk with *lncIrf8/Irf8* both in vitro and in vivo will be subject of our future work and described elsewhere.

Reviewer #2 (Recommendations for the authors):I find the quality and interpretation of the genomics data to be excellent and I have no major concerns with that data. I do have some specific comments around the interpretation of the cellular data and the model presented.Specific comments1. The authors conclude that lncIRF8 promoter KO hematopoietic progenitors are not capable of differentiating to pDCs (Figure 2C, Figure 2 Suppl 3D). However, it is now understood that Irf8 is not essential for pDC development (see Sichien Immunity 2016 where they study WT:IRF8-/- BM chimeras) and that Irf8 deregulates some key markers, including CD11b. The authors need to account for this discrepancy. Are pDCs with aberrant marker expression produced in the lncIRF8 promoter KO, and if not, why does this phenotype differ from the Irf8 KO?

We appreciate the point made by the reviewer. In our study on DC differentiation of *lncIrf8* promoter KO in vitro and following transplantation into sublethally irradiated mice in vivo*,* we observed low to absent CD11c+ CD11b- B220+ pDC (Figure 2C, Figure 2—figure supplement 3) or Gr1- Siglec H^+^ pDC (Figure 3A and D, G and J), respectively.

We know the work by Sichien et al., 2016 on WT:*Irf8-/-* BM chimeras and the conclusions reached from their work that IRF8 is dispensable for pDC development. The results in Sichien et al., 2016 are based on WT:*Irf8-/-* BM chimeras upon transplantation in lethally irradiated mice and thus the divergent results compared to the classical *Irf8-/-* mice might be due to the experimental set-up WT:*Irf8-/-* BM chimeras versus *Irf8* knockout mice (see also our response above to Essential revisions 2).

We now followed the advice of the reviewer and checked whether the *lncIrf8* promoter KO had an impact on DC marker genes, such as CD11c and CD11b. We visualized CD11c and CD11b expression (gating on single cells) on day 7 and 9 of Flt3L directed DC differentiation in *lncIrf8* promoter KO and control cells. In *lncIrf8* promoter KO cells CD11c expression was slightly reduced and CD11b expression was slightly increased (see above Author response image 2 and our response to Essential revisions 2).

This indicates that *lncIrf8* promoter KO does not have a major effect on CD11c and CD11b expression. Given this negative result we prefer not to include these data in the manuscript.

2. There does not appear to be compelling evidence that IRF8 binds in the lncIRF8 promoter in cDC1. Instead, IRF8 binds in the more 5' part of the +32kb enhancer identified by Durai 2019. How then is a putative IRF8 repressor complex linked to the lncIRF8 promoter as the authors propose? Moreover, the model does not appear to address why the IRF8 repressor complex binding to the IRF8 repressor complex would be active in cDC1 and not pDC.

The point made by the reviewer is well taken. In our model we proposed that binding of the putative IRF8 repressor to the +32 kb enhancer is mainly in the 5’ part of the +32 kb enhancer, as demonstrated by the ChIP-seq of IRF8. There is a strong IRF8 binding signal detected in cDC and weak to no IRF8 binding signal in pDC (Figure 1A, Figure 1—figure supplement 1, Figure 2—figure supplement 2 and Figure 7B, ChIP-seq IRF8 signal).

Thus, we propose the putative IRF8 repressor complex is only active in cDC and not pDC (Figure 7B). We agree with the reviewer that at this stage of analysis we do not know why IRF8 binds to the 5’ part of the +32 kb enhancer in cDC but not in pDC. Addressing this point should be subject for further studies. See also our response above to Essential revisions 3.

3. The authors use 2 versions of the HoxB8 MPP differentiation system (lines 207-234). Firstly, it is unclear, why two similar approaches are used, but more importantly, the second system outlined in figure 2 Suppl 3 shows that the lncIRF8 promoter is also required for cDC2 differentiation. This is a concern, as IRF8 is not expressed in cDC2, and Irf8-/- mice produce cDC2.

In the present study we used two approaches to differentiate the *lncIrf8* promoter KO cells, thereby following the DC differentiation systems described in Xu et al., *Eur. J. Immunol.*, 2022:

The first system is spontaneous DC differentiation, where we just remove estrogen (E2) and keep SCF, Flt3L (low concentration), hyper-IL6 and IGF1 in culture. Under these conditions most of the *lncIrf8* promoter KO cells moved to Gr1+ monocytes (Figure 2E). This spontaneous DC differentiation mimics in vitro the differentiation potential of these cells in vivo upon cell transplantation, where most of the *lncIrf8* promoter KO cells also differentiated into Gr1+ monocytes (Figure 3B and H).

The second system is Flt3L directed DC differentiation, where estrogen (E2) and the growth promoting cytokines SCF, Flt3L (low concentration), hyper-IL6 and IGF1 are removed and high Flt3L concentration is applied to direct HoxB8 MPP differentiation towards DC, and all DC subsets (cDC1, cDC2 and pDC) are obtained. In the manuscript we used both approaches, spontaneous DC differentiation (-E2) and Flt3L directed DC differentiation, to support the conclusions reached.

Concerning the reduced cDC2 population in Figure 2—figure supplement 3G, this is because the cDC2 population was gated from CD11c+ cells and calculated based on living single cells. In the *lncIrf8* promoter KO most cells differentiated into Gr1+ monocytes (Figure 2E) and thus CD11c+ cDC2 were much lower in the *lncIrf8* promoter KO compared to control.

We now also calculated cDC2 based on CD11c+ cells and show these data as new Figure 2—figure supplement 3H. We did not observe significant differences between *lncIrf8* promoter KO and control cells on day 8 and 10. On day 6 the spontaneous DC differentiation of control cells progressed slower than the *lncIrf8* promoter KO cells, thus lower frequency of cDC2 was observed. We added this information to the legend for Figure 2—figure supplement 3 (page 34, lines 1214-1217).